

# Technical Note: TimeFRAME - A Bayesian Mixing Model to Unravel Isotopic Data and Quantify Trace Gas Production and Consumption Pathways for Timeseries Data

Eliza Harris[1,*], Philipp Fischer[1,2,*], Maciej P. Lewicki[3], Dominika Lewicka-Szczebak[4], Stephen J. Harris[5,6], and Fernando Perez-Cruz[1]

[1]Swiss Data Science Center, ETH Zürich and EPFL, Lausanne, Switzerland
[2]Now at: BSI Business Systems Integration AG, 5405 Baden
[3]Institute of Nuclear Physics, Polish Academy of Sciences, Krakow, Poland
[4]Institute of Geological Sciences, University of Wrocław, Wrocław, Poland
[5]Australian Nuclear Science and Technology Organisation, Lucas Heights, NSW, 2234, Australia
[6]School of Biological, Earth and Environmental Sciences, UNSW Sydney, NSW, 2052, Australia
[*]These authors contributed equally to this work.
**Correspondence:** Eliza Harris (eliza.harris@ethz.sdsc.ch)

**Abstract.** Isotopic measurements of trace gases such as $N_2O$, $CO_2$ and $CH_4$ contain valuable information about production and consumption pathways. Quantification of the underlying pathways contributing to variability in isotopic timeseries can provide answers to key scientific questions, such as the contribution of nitrification and denitrification to $N_2O$ emissions under different environmental conditions, or the drivers of multiyear variability in atmospheric $CH_4$ growth rate. However, there is

currently no data analysis package available to solve isotopic production, mixing and consumption problems for timeseries data in a unified manner while accounting for uncertainty in measurements and model parameters. Bayesian hierarchical models combine the use of expert information with measured data and a mathematical mixing model while considering and updating the uncertainties involved, and are an ideal basis to approach this problem.

Here we present the `TimeFRAME` data analysis package for 'Time-resolved Fractionation And Mixing Evaluation'. We

use four different classes of Bayesian hierarchical model to solve production, mixing and consumption contributions using multi-isotope timeseries measurements: i) independent time step models, ii) Gaussian process priors on measurements, iii) Dirichlet-Gaussian process priors, and iv) generalized linear models with spline bases. All four models have been extensively tested in different variations and for a multitude of scenarios. Dirichlet-Gaussian process prior models have been found to be most reliable, allowing for simultaneous estimation of hyperparameters via Bayesian hierarchical modeling. Generalized

linear models with spline bases seem promising as well, especially for fractionation estimation, although the robustness to real datasets is difficult to assess given their high flexibility. Experiments with simulated data for $\delta^{15}N^{bulk}$ and $\delta^{15}N^{SP}$ of $N_2O$ showed that model performance across all classes could be greatly improved by reducing uncertainty in model input data - particularly isotopic endmembers and fractionation factors. The addition of the $\delta^{18}O$ additional isotopic dimension yielded a comparatively small benefit for $N_2O$ production pathways but improved quantification of the fraction of $N_2O$ consumed.



The `TimeFRAME` package can be used to evaluate both static and timeseries datasets, with flexible choice of the number and type of isotopic endmembers and the model set up allowing simple implementation for different trace gases. The package is available in R, and is implemented using `Stan` for parameter estimation, in addition to supplementary functions re-implementing some of the surveyed isotope analysis techniques.

# 1  Introduction

Analysis of isotopic signatures is frequently used in environmental sciences to infer production and consumption pathways for trace gases. For example, $N_2O$ isotopic composition reflects the production via different pathways (including microbial denitrification, nitrification, fungal denitrification), mixing within the soil airspace, and consumption via complete denitrification. The different production pathways have distinct isotopic 'endmembers', which describe the isotopic composition of emitted $N_2O$. Following emission, $N_2O$ from different pathways mixes in the soil airspace, described using the mixing equation

(Ostrom et al., 2007; Fischer, 2023):

$$\delta_{\mathrm{mix}} = \sum_{k=1}^{K} f_k \delta_k \tag{1}$$

where $\delta_{\mathrm{mix}}$ is the isotopic composition of a mixture of two or more sources enumerated by $k = 1, ..., K$ with isotopic compositions designated $\delta_k$ and fractional contributions to the mixture designated by $f_k$. This equation assumes that the light isotope has a much greater concentration than the heavy isotope, which is valid for common trace gases such as $CO_2$, $CH_4$ and $N_2O$.

$N_2O$ is consumed during complete denitrification to $N_2$, which favours the light isotope and thus leads to progressive enrichment of the remaining $N_2O$ pool. The isotopic effect of consumption can be approximated using the Rayleigh equation (Mariotti et al., 1981; Ostrom et al., 2007; Fischer, 2023):

$$\delta_{\mathrm{substr,r}} = \delta_{\mathrm{substr,r=1}} + \epsilon \ln(r) \tag{2}$$

where $\delta_{\mathrm{substr,r=1}}$ and $\delta_{\mathrm{substr,r}}$ are the isotopic composition of the substrate being consumed before consumption ($r = 1$) and

when a certain fraction ($r - 1$) has been consumed, $\epsilon$ is the fractionation factor for the reaction in permil (‰) and $r$ is the fraction of substrate remaining where $r = 0$ represents a complete reaction.

We can combine equations 1 and 2 for a full model of mixing and fractionation of the subsequent mixture; for example, mixing of $N_2O$ from different sources within the soil airspace, followed by complete reduction of a certain fraction of $N_2O$, before measurement of $N_2O$ isotopic composition (Fischer, 2023):

$$\delta = \sum_{k=1}^{K} f_k \delta_k + \epsilon \ln(r) \tag{3}$$

where $\delta$ is the measured isotopic composition. In this equation, we assume that mixing occurs before fractionation, when in reality mixing and fractionation are likely occurring simultaneously depending on the soil pore size distribution and connectivity, the availability of different substrates, and the microbial community present (Denk et al., 2017; Yu et al., 2020; Lewicka-Szczebak et al., 2020). Further uncertainty in the model equation relates to open vs. closed system fractionation, describing



renewal of the $N_2O$ pool relative to the rate of $N_2O$ consumption (Yu et al., 2020; Lewicka-Szczebak et al., 2020). However, the
largest uncertainties in evaluation of this equation to interpret the measured isotopic composition $\delta$ relate to the endmembers
for different sources $\delta_k$ and the fractionation factor $\epsilon$.

A commonly used approach to interpret trace gas isotopic measurements is the application of dual-isotope mapping, which
utilises the relationship between two isotopic parameters to infer pathways, for example $\delta^{15}N^{bulk}$ and $\delta^{15}N^{SP}$ in the case

of $N_2O$. The mapping approach can be used to roughly estimate the dominance of different pathways and the importance of
fractionation during consumption (Wolf et al., 2015; Lewicka-Szczebak et al., 2017; Wu et al., 2019; Yu et al., 2020; Rohe et al.,
2021). However, they fail to provide quantitative determination of different pathways or to estimate uncertainty for individual
samples. Moreover, mapping approaches are limited to mixing scenarios involving only two sources, which - for example -
does not allow for the differentiation of contributions from the nitrification and fungal denitrification pathways which have

similar $\delta^{15}N^{SP}$ signatures. In addition, there are no statistical packages available to implement these mapping approaches,
calling into question the reproducibility among studies using this approach.

Bayesian approaches to solve isotopic mixing models have been successfully implemented in several well-known frameworks
(R packages `MixSIAR`, `simmr`) (Parnell et al., 2013; Stock et al., 2018). These advanced models are used to resolve the
contribution of multiple sources to a mixture using a range of Bayesian statistical techniques, and are widely used for applications

such as animal diet partitioning (Stock et al., 2018). However, these packages do not offer the capability to deal with pool
consumption and Rayleigh fractionation, and thus are not suitable for the interpretation of trace gas isotopic measurements
where consumption/destruction plays a key role (Fischer, 2023; Lewicka-Szczebak et al., 2020).

The FRAME (Fractionation And Mixing Evaluation) model provided the first Bayesian tool to include both mixing and
fractionation for the interpretation of isotopic data (Lewicka-Szczebak et al., 2020; Lewicki et al., 2022). FRAME applies a

Markov-Chain Monte Carlo (MCMC) algorithm to estimate the contribution of individual sources and processes, as well as
the probability distributions of the calculated results (Lewicki et al., 2022). However, the FRAME model can only be applied
to timeseries data by solving separately for single or aggregated points. Although the contribution of different pathways may
vary strongly on short timescales, model parameters - such as isotopic endmembers and fractionation factors - are expected to
show minimal variability between subsequent points in a time series.

Here we present the TimeFRAME extension to the FRAME model to allow for efficient analysis of timeseries data.
TimeFRAME uses four classes of model to incorporate time series information: i) independent time step models, ii) Gaussian
process priors on measurements, iii) Dirichlet-Gaussian process priors, and iv) generalized linear models with spline bases.
The models are solved for the contribution of different pathways, endmembers and fractionation factors within a MCMC
framework, and the full posterior distributions of parameters are reported. The isotopes, endmembers, fractionation factors and

model set up are defined by the user, allowing flexible application to many isotopic problems.





## 2 Methodology

### 2.1 Inference of Source Contributions

One objective of studying isotopic signatures is to determine the source contributions $f_1 \cdots f_K$ from measurements of the mixture. However, measuring one single isotopic species will only be efficient in distinguishing between a maximum of two sources. For additional sources, or if consumption of the mixture needs to be accounted for, multiple isotopic species are necessary. Analysis of $N_2O$ sources and pathways, for instance, can include analysis of $\delta^{15}N^{\mathrm{bulk}}$, $\delta^{15}N^{\mathrm{SP}}$ and $\delta^{18}O$. The vector of $d$ different isotopic species shall be denoted by $X \in \mathbb{R}_d$. Measurements of the isotopic endmember for each individual source enumerated by $k = 1, ..., K$ are assumed to be known and denoted by $S_1, ..., S_K \in \mathbb{R}_d$ together with the fractionation factor $\epsilon \in \mathbb{R}_d$. Using vector and matrix notation they can later be used to state the mixing equation in vector form:

$$\mathbf{f} := [f_1 \cdots f_K]^T \in \mathbb{R}^K \tag{4}$$

$$\mathbf{S} := [S_1 \cdots S_K]^T \in \mathbb{R}^{d \times K} \tag{5}$$

The case of Rayleigh fractionation as expressed in Eq. 3 can be similarly expressed in vectorised form:

$$X = \mu(\mathbf{f}, r) := \mathbf{S}\mathbf{f} + \epsilon \ln(r) \tag{6}$$

In a simple example with two sources and measurements of two isotopic species $K = d = 2$ the mixing equation can be solved (assuming convergence is possible) for the parameters of interest using linear algebra (see (Fischer, 2023) for details). More generally, mixing and fractionation according to Eq. 6 can be solved for an arbitrary number of sources as long as an equal number of isotopic species is available, ie. $K = d \geq 2$. In this case the linear system of equations can be written in matrix terms, and augmented with the sum constraint on $\mathbf{f}$:

$$\tilde{X} := \begin{bmatrix} X \\ 1 \end{bmatrix} = \begin{bmatrix} \mathbf{S} & -\epsilon \\ 1^{\mathrm{T}} & 0 \end{bmatrix} \begin{bmatrix} \mathbf{f} \\ \ln(r) \end{bmatrix} =: \tilde{\mathbf{S}}\tilde{\mathbf{f}} \tag{7}$$

This $d + 1$-dimensional linear system of equations can be addressed with decomposition techniques and its solution can be expressed as $\tilde{\mathbf{f}} = \tilde{\mathbf{S}}^{-1}\tilde{X}$. A unique solution exists if $\tilde{\mathbf{S}}$ is invertible, or equivalently, if none of the mixing lines as well as the consumption line are co-linear. Only non-negative solutions $\tilde{\mathbf{f}} \geq 0$ are feasible to ensure that the source contributions $f$ correspond to mixing weights and that $0 < r \leq 1$.

A flaw of the isotope mapping approach as presented above is that it does not take measurement uncertainty into account. However, this can easily be added by formulating the measurements $X$ as random variables with expected value given by the mixing equation $\mathbb{E}[X] = \mu(\mathbf{f}, r)$. Most commonly, measurements are modeled using the Gaussian distribution with independent components and variance $\eta^2 \in \mathbb{R}$, thus allowing the mixing-fractionation equation to be stated as:

$$X \sim \mathcal{N}_d(\mu(\mathbf{f}, r), \eta^2 \mathbb{1}) = \mathcal{N}_d(\mathbf{S}\mathbf{f} + \epsilon \ln(r), \eta^2 \mathbb{1}) \tag{8}$$



A maximum likelihood solution to this mixing-fractionation equation can be pursued (see (Fischer, 2023) for details), however

this limits the framework to parameters that can be approximated with a Gaussian distribution. Often, the epistemic uncertainty

of source isotopic endmembers is modelled as a uniform distribution to best account for the combination of measurement

uncertainty and true variability in endmember values (Lewicki et al., 2022). Bayesian statistics is useful to incorporate all

assumptions and constraints into the model, as well as to employ numerical inference methods for source contribution estimation

and uncertainty estimation.

## 2.2    Stationary inference

Stationary inference involves inference for the source contributions $\mathbf{f} \in \mathcal{S}_K$, where $\mathcal{S}_K$ is the K-simplex, and the fraction of the

pool remaining $r \in [0,1]$ for one single measurement independent of time. This can be accomplished with the original FRAME

model (Lewicki et al., 2022). FRAME constructs a prior and likelihood structure where the isotopic species measurements

$X \in \mathbb{R}^d$ are independently normally distributed with variance vector $\eta^2 \in \mathbb{R}_+^d$ around a mean given by an arbitrary mixing

equation $\mu(\mathbf{f}, r)$. The source contributions $\mathbf{f}$ are then equipped with a flat Dirichlet prior and the fraction remaining $r$ with a

uniform prior:

$$\mathbf{f} \sim \mathrm{Dir}(1), \ r \sim \mathrm{Uni}(0,1)$$
$$X|\mathbf{f}, r \sim \mathcal{N}_d(\mu(\mathbf{f}, r), \eta^2) \tag{9}$$

The auxiliary parameters for the mixing equation $\mathbf{S}$ and $\epsilon$ are understood to be random variables as well, with predetermined

fixed priors that are omitted from the model description above. Choosing those priors is dependent on the origin of the data and

thus not subject to the further engineering of inference models in the following sections. The likelihood of $X$ is understood to

be implicitly conditional on these auxiliary parameters. This means that a joint posterior $\pi(\mathbf{f}, r, \mathbf{S}, \epsilon | X)$ is fit by the model and

the reported posterior $\pi(\mathbf{f}, r | X)$ is simply its marginalization.

     The FRAME model can be extended by taking different choices of prior distributions for the parameters of interest (the

source contributions $\mathbf{f}$ and the fraction remaining $r$). The Jeffreys prior for source contributions is constructed by computing

the Fisher information matrix and choosing the probability distribution proportional to the square root of its determinant. For

the source contributions $\mathbf{f} = \begin{pmatrix} 1-f \\ f \end{pmatrix} \in \mathcal{S}_2$ of two sources $S_1, S_2 \in \mathbb{R}$ the computation can be done by omitting the influence

of fractionation, leading to a uniform prior over the domain of $\mathbf{f}$. For multiple source contributions this is equivalent to the flat

Dirichlet distribution used in the original FRAME model.

Taking now fraction remaining with regards to Rayleigh fractionation $r \in [0,1]$ independently of the mixing weights $\mathbf{f}$

into account, the Jeffreys prior can be computed relative to the pure mixing solution $M = X - S_1(1-f) - S_2 f \in \mathbb{R}$ with

fractionation factor $\epsilon \in \mathbb{R}$ (Fischer, 2023):

$$\mathcal{I}_r(r) \propto -\mathbb{E}\left[ \frac{d^2 r}{dr^2} \frac{(M - \epsilon \ln(r))^2}{\eta^2} \Big| r \right] = \frac{2\epsilon^2}{\eta^2 r^2} \propto \frac{1}{r^2} \tag{10}$$

Therefore the objective prior for the fraction remaining $r$ is given by $\pi(r) \propto \frac{1}{r}$ for $r \in [0,1]$, which is also known as the

logarithmic prior; since it cannot be normalized it is an improper prior. Additionally, even though this prior can be considered



uninformative for $r$ individually according to the Jeffreys criterion, a joint prior could lead to different results, although priors are typically chosen as independent distributions.

In the case of Rayleigh fractionation, however, it might be more reasonable to use a different distribution that is somewhere in between the uniform and logarithmic prior and incorporates the bounds to the interval $[0,1]$ as well, which is a similar idea
to prior averaging (Berger et al., 2015). The beta distribution offers a functional form that is similar to the Jeffreys prior, but can be normalized. Parameterizing the distribution with a restricted concentration parameter $\alpha \in [0,1]$, the form $\text{Beta}(\alpha, 1)$ is the uniform distribution for $\alpha = 1$ and converges to the Jeffreys prior for $\alpha \to 0$, thus expressing a generalized approach.

## 2.3 Time series inference

In order to incorporate time series information in the inference procedure, the model can be extended to work with multiple
measurements at different points in time. The source contribution $\mathbf{f}$ and fraction remaining $r$ are assumed to be functions with respect to time $\mathbf{f}_\tau$ and $r_\tau$ and the measurements correspond to samples in time $X_t = X(\tau_t)$ at $n$ discrete time points $\tau_1, ..., \tau_n$.

Now the measurements can be grouped into a measurement matrix $\mathbf{X} := [X_1 \ldots X_N] \in \mathbb{R}^{d \times N}$, where the time dimension is along the matrix rows. Inference of the parameters can be done at the identical time points $\mathbf{f}_t = \mathbf{f}(\tau_t)$ and $r_t = r(\tau_t)$, so that they can be grouped into similar matrices as well: $\mathbf{F} := [\mathbf{f}_1 \ldots \mathbf{f}_N] \in \mathbb{R}^{K \times N}$ and $\mathbf{r} := [r_1 \ldots r_N] \in \mathbb{R}^{1 \times N}$. This grouping has
the advantage that the mixing equation can be expressed in vectorized form over all time points without changing its general layout (Fischer, 2023):

$$\mathbb{E}[\mathbf{X}|\mathbf{F},\mathbf{r}] = \mu(\mathbf{F},\mathbf{r}) = \mathbf{S}\mathbf{F} + \epsilon \ln(\mathbf{r}) \tag{11}$$

### 2.3.1 Independent time steps

The simplest method to extend the stationary model is to assume complete independence between all points in time. This
reduces the time series problem to a set of $N$ independent stationary problems with one single measurement point each, thus the same stationary FRAME model can be used for each point. The vector of measurement errors $\eta \in \mathbb{R}^d$ is now also allowed to vary in time as $\eta_1, ..., \eta_N$

$$\mathbf{f}_t \sim \text{Dir}(1), \ r_t \sim \text{Uni}(0,1) \quad \forall t$$
$$X_t|\mathbf{f}_t, r_t \sim \mathcal{N}_d(\mu(\mathbf{f}_t, r_t), \eta_t^2) \quad \forall t \tag{12}$$

The prior on the series of source contributions $\mathbf{f}_t$ and pool fraction remaining $r_t$ is now fully independent in time and the
information that could be contained in the fact that some measurements are closer in time than others is ignored.

Prior information can be encoded into the prior distribution for $\mathbf{f}_t$ by introducing a concentration parameter $\sigma \in \mathbb{R}_+^K$ as well as a parameter $\alpha \in (0,1)$ that interpolates between the uniform prior for $r_t$ and the Jeffreys prior using the beta distribution as described in Section 2.2. This allows for the inclusion of information that is universally true for all time points simultaneously. If no information is available the model can be extended by adding an additional hierarchical layer for these parameters with
weakly informative hyperpriors being the gamma distribution $\Gamma(2,2)$ on the positive real axis for concentrations $\sigma$ and the



uniform $\mathrm{Uni}(0,1)$ for $\alpha$:

$$\sigma \sim \Gamma(2,2), \alpha \sim \mathrm{Uni}(0,1)$$
$$\mathbf{f}_t \sim \mathrm{Dir}(\sigma), \, r_t \sim \mathrm{B}(\alpha,1) \qquad \forall t \tag{13}$$
$$X_t | \mathbf{f}_t, r_t \sim \mathcal{N}_d(\mu(\mathbf{f}_t, r_t), \eta_t^2) \quad \forall t$$

### 2.3.2 Gaussian Process priors

Time series information can be incorporated by various methods in the case of unconstrained random variables. An obvious
method is to apply the FRAME model with independent time steps on a preprocessed measurement series. The timeseries preprocessing can be done before model application, without consideration of the Bayesian mixing model. Candidate preprocessing algorithms are kernel smoothing, spline smoothing, and local polynomial regression. While the latter can offer uncertainty estimates of the smoothed time series, a holistic treatment of estimation with uncertainty can be offered by Gaussian process regression.

Despite the possibility of running the simple time-independent model on preprocessed measurement time series, it is beneficial to combine both steps into an advanced model; for example, the problem specific geometry could influence the feasibility of a region in measurement space. A combined model will include a Gaussian process prior on the measurements $X_t$ such that posterior means $W_t$ can be estimated and used to drive the FRAME model with independent time steps (Section 12). The Gaussian process is shifted and scaled to align with the empirical mean $\hat{\mu}_X$ and standard deviation $\hat{\sigma}_X$ of the measurements
$X_t$ and controlled by a kernel function $\mathbf{G}$ (Fischer, 2023):

$$\mathbf{f}_t \sim \mathrm{Dir}(1), \, r_t \sim \mathrm{Uni}(0,1) \qquad \forall t$$
$$W_t | \mathbf{f}_t, r_t \sim \mathcal{N}_d(\mu(\mathbf{f}_t, r_t), \frac{\eta_t^2}{2}) \quad \forall t$$
$$W \sim \mathcal{GP}_d(\hat{\mu}_X, \hat{\sigma}_X, \mathbf{G}) \tag{14}$$
$$X_t | W_t, \mathbf{f}_t, r_t \sim \mathcal{N}_d(W_t, \frac{\eta_t^2}{2}) \qquad \forall t$$

The distribution on the latent estimates $W_t$ is the product of the Gaussian process prior as well as the independent normal distribution around the mixing estimate. Ideally, this model does not need to include sampling of $\mathbf{f}_t$ and $r_t$ because if the mixing equation can be expressed as a linear system of equations (Eq. 7) then the smooth measurement series $W_t$ is sufficient
to solve for the source contribution and fractionation parameters directly. In practice, this approach reduces to applying isotopic mapping techniques to the time series that is preprocessed using Gaussian process smoothing.

If the mixing equation is not explicitly inverted but evaluated by sampling the parameters $\mathbf{f}_t$ and $r_t$, then the latent variables $W$ can be marginalized over and eliminated from the model. The product density of $W$ can also be expressed using known identities (Pedersen and Petersen, 2012) for each separate isotopic measurement dimension $j = 1, ..., d$ in terms of its empirical
mean $\hat{\mu}_{X,j}$, empirical standard deviation $\hat{\sigma}_{X,j}$ and noise variance $\eta_j^2$. Using the Cholesky decomposition these distribution




parameters can efficiently be computed and used for sampling, thus the latent parameters $W$ can be eliminated from the model and the likelihood of each row $X_{j:}$ can be directly computed (Fischer, 2023):

$$\mathbf{F} \sim \text{Dir}(1), \; \mathbf{r} \sim \text{Uni}(0,1)$$

$$\mu(\mathbf{F},\mathbf{r}) \sim \mathcal{GP}_d(\hat{\mu}_X, \hat{\sigma}_X, \mathbf{G}) \tag{15}$$

$$\mathbf{X}_{j:}^T|\mathbf{F},\mathbf{r} \sim \mathcal{N}_N(\tilde{\mu}_j, \tilde{\Sigma}_j + \frac{\eta_t^2}{2}\mathbb{1}) \quad \forall j = 1,...,d$$

Gaussian process priors on measurements use only one single hyperparameter which is the correlation length $\rho$ used to compute the kernel matrix $\mathbf{G}_{ij} = \kappa_\rho(\tau_i, \tau_j)$. The scale of the Gaussian process is always set to the empirical standard deviation of the data and is thus considered fixed. In order to compile a fully hierarchical Bayesian model an inverse gamma distribution $\frac{1}{\rho} \sim \Gamma(2,2)$ can be used as hyperprior for the correlation length, assuming that the time scales are properly normalized.

### 2.3.3 Dirichlet-Gaussian Process priors

To make use of time series information in the source contributions $\mathbf{f}$ and the fraction reacted $r$, direct priors are desired. These priors can be constructed by sampling auxiliary variables from multiple independent Gaussian processes $\mathbf{Z} \sim \mathcal{GP}_K(\mathbf{G})$ and at each point in time inverting the log ratio transformations on the simplex in order to create a time series of simplex-valued variables $\mathbf{f}_t$ . The fraction reacted $r_t$ is constrained to the interval $[0,1]$ and can thus be linked for instance by applying the logit transform $\text{logit}(r) = \ln\frac{r}{1-r}$ at each point in time, which maps it to the entire real axis. Hyperparameters for correlation length $\rho \in \mathbb{R}_+$ and concentration $\sigma \in \mathbb{R}_+$ are used to compute the kernel matrix $\mathbf{G}_{ij} = \sigma^2\kappa_\rho(\tau_i, \tau_j)$ for the Gaussian process. The general shape of these priors is visualized in Figure 1. Working with the matrix of source contributions $\mathbf{F} = [\mathbf{f}_1,...,\mathbf{f}_n] \in \mathbb{R}^{K \times N}$ and of the fraction reacted $\mathbf{r} = [r_1,...,r_n] \in \mathbb{R}^{1 \times N}$ the model can be stated in vectorized form, where the link functions are understood to be column-wise (Fischer, 2023):

$$\text{clr}(\mathbf{F}) \sim \mathcal{GP}_K(\mathbf{G})$$

$$\text{logit}(\mathbf{r}) \sim \mathcal{GP}(\mathbf{G}) \tag{16}$$

$$X_t|\mathbf{f}_t, r_t \sim \mathcal{N}_d(\mu(\mathbf{f}_t, r_t), \eta_t^2) \quad \forall t$$

Both link functions used can easily be inverted once random variables $\mathbf{Z} \sim \mathcal{GP}_K(\mathbf{G})$ and $\mathbf{Y} \sim \mathcal{GP}(\mathbf{G})$ are sampled from Gaussian processes over time $t = 1,...,N$. The inverse of the CLR transform is given by the softmax function and the inverse of the logit link is given by the sigmoid function (see Fischer (2023)). The prior on the source contribution parameters $\mathbf{G}$ is known as a generalized Gaussian process prior, and techniques such as Taylor expansion can be used to derive analytic approximations (Chan and Dong, 2011). Its marginal is a softmax transformed multivariate Gaussian, which is also known as a logistic normal distribution and serves as an approximation to the Dirichlet distribution (Aitchison and Shen, 1980; Devroye, 1986; Fischer, 2023). Mapping using centered log ratio (CLR) transforms thus creates a time series of random variables with approximate Dirichlet marginals, which is referred to as a *Dirichlet-Gaussian process* (DGP) (Chan, 2013).

The marginals are controlled by the parameter $\sigma$ of the Gaussian process, which now acts as the concentration parameter of the Dirichlet distribution. Since the covariance kernel $\mathbf{G}$ is scaled to generate Gaussian random variables with unit variance





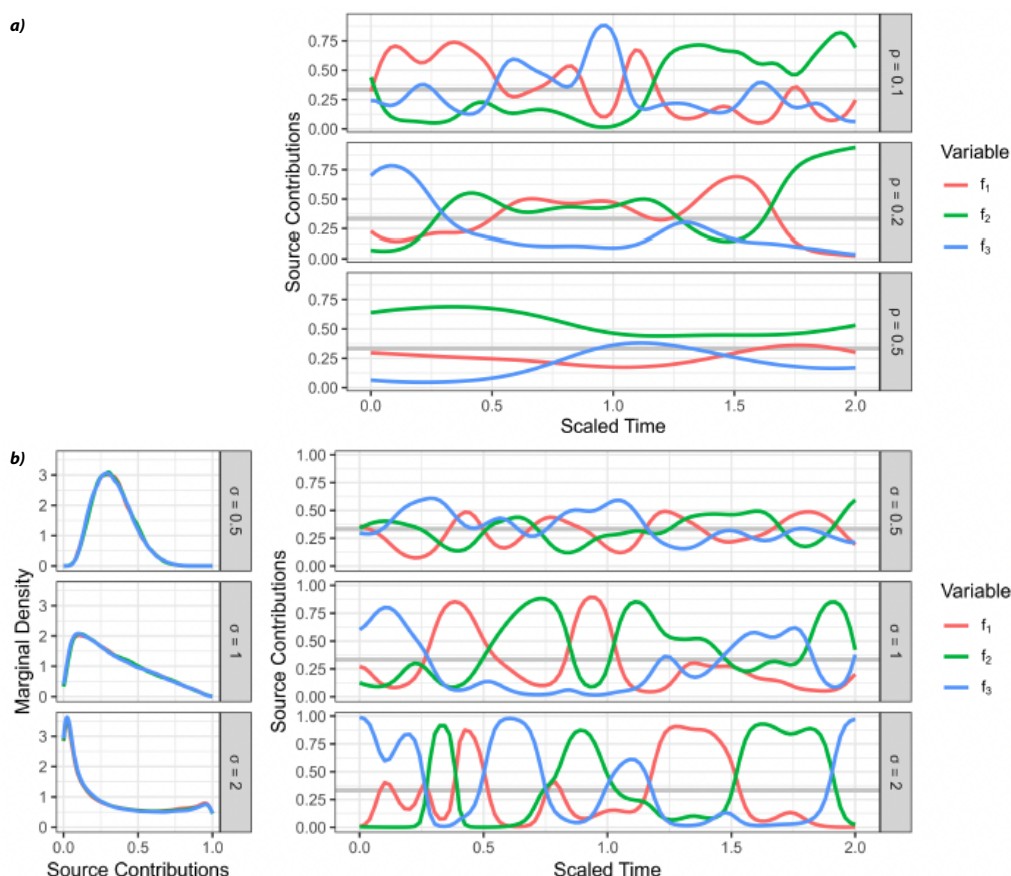

**Figure 1.** Examples of Generalized Gaussian Process prior with the radial basis function kernel using different values for correlation length ($\rho$) and concentration ($\sigma$). $a$) Prior observation for three sources ($f_1$, $f_2$, $f_3$) mapped to the simplex using the centred log ratio transform, shown over an arbitrary time axis using different values of $\rho$ with $\sigma = 1$. $b$) Estimated marginal densities (LHS) of transformed Gaussian process priors for different values of the concentration parameters $\sigma$ with $\rho = 0.1$, with a prior observation for three sources ($f_1$, $f_2$, $f_3$) shown over an arbitrary time axis on the RHS.

if $\sigma = 1$, the marginal distribution in that case is approximately the uniform $\mathrm{Dir}(1)$. This can be seen by sampling from the

225    generalized Gaussian process priors and estimating the marginals, as shown in Figure 1$b$.

Using the isometric log ratio transform ILR instead of CLR reduces the number of Gaussian processes that need to be sampled to $K - 1$ for the source contributions. Inverting this link function is accomplished by applying an orthonormal base transform $\mathbf{U}$ to the random variables, and then applying the softmax function. Since interpretability of the sampled Gaussian process variables is not required, any orthonormal basis is suitable and a simple construction using Gram-Schmidt orthogonalization is





chosen (Nesrstová et al., 2022; Fischer, 2023):

$$\text{ILR}(\mathbf{F}) \sim \mathcal{GP}_{K-1}(\mathbf{G})$$

$$\text{logit}(\mathbf{r}) \sim \mathcal{GP}(\mathbf{G})$$

$$X_t | \mathbf{f}_t, r_t \sim \mathcal{N}_d(\mu(\mathbf{f}_t, r_t), \eta_t^2) \quad \forall t$$

(17)

Shorthand notation $\mathcal{GP}(\mathbf{G}) = \mathcal{GP}(0, 1, \mathbf{G})$ is used with correlation length $\rho$ and scale $\sigma$ included in the kernel computation $\mathbf{G}_{ij} = \sigma^2 \kappa_\rho(\tau_i, \tau_j)$. Both kernel parameters $\rho$ and $\sigma$ can be set in advance or given weak hyperpriors. The inverse gamma distribution $\frac{1}{\rho} \sim \Gamma(2, 2)$ and the regular gamma distribution $\sigma \sim \Gamma(2, 2)$ are chosen under the assumption that the time variables $\tau_1, ..., \tau_N$ are scaled appropriately. This hierarchical model benefits especially from the reduced number of Gaussian processes sampled when using the ILR transform, since the kernel covariance matrix must be reconstructed in every sampling step. The number of hyperparameters could be increased by using separate concentrations and correlation lengths for the source contributions $\mathbf{F}$ and the fraction remaining $\mathbf{r}$.

### 2.3.4 Spline-based priors

An alternative to Gaussian process priors are spline basis functions, which can be used to construct a linear fitting operation that is then mapped to simplex space. This allows for the addition of exogenous variables as predictors of source contributions or fractionation. A cubic spline basis of $M$ basis functions (Figure 2) is evaluated at the measurement points $\tau_1, ..., \tau_N$ to form the evaluation matrix $\mathbf{H} \in \mathbb{R}^{N \times M}$ with $\mathbf{H}_{ij} = s_j(\tau_i)$ for polynomial basis functions $s_1(\cdot), ..., s_M(\cdot)$. The time series of source contributions in simplex space is reconstructed with the basis coefficients $\mathbf{b}_k \in \mathbb{R}^M$ for each source $k = 1, ..., K$ arranged to the matrix $[\mathbf{b}_1 \cdots \mathbf{b}_K]^T = \mathbf{B} \in \mathbb{R}^{L \times M}$ and coefficients for fractionation $\mathbf{c} \in \mathbb{R}^{1 \times M}$. This type of model is therefore part of the generalized linear model class (Nelder and Wedderburn, 1972) and allows for easy extension with fixed effects relating to measurement dimensions as well as random effects for experiment replication. It will thus further be referred to as generalized linear model with spline basis (spline GLM) (Fischer, 2023):

$$\mathbf{B}, \mathbf{c} \sim \mathcal{N}(0, 1)$$

$$\text{clr}(\mathbf{F}) = \mathbf{B}\mathbf{H}^T$$

$$\text{logit}(\mathbf{r}) = \mathbf{c}\mathbf{H}^T$$

$$X_t | \mathbf{f}_t, r_t \sim \mathcal{N}_d(\mu(\mathbf{f}_t, r_t), \eta_t^2) \quad \forall t$$

(18)

In consequence, the distribution of the basis coefficient vector $\mathbf{b}_k = \mathbf{B}_{k:}^T \in \mathbb{R}^M$ before transformation has distribution $\mathbf{b}_k \sim \mathcal{N}_M(0, \mathbb{1})$ for source $k = 1, ..., K$. After application of the spline basis transform it is thus still Gaussian $\mathbf{H}\mathbf{b}_k \sim \mathcal{N}_N(0, \mathbf{H}\mathbf{H}^T)$, although with a modified covariance matrix $\mathbf{H}\mathbf{H}^T \in \mathbb{R}^{N \times N}$. Since the inverse centered log ratio transform maps Gaussian random variables with unit variance approximately to a uniform Dirichlet distribution, it makes sense to scale the basis transform such that $\frac{1}{N} \text{Tr}(\mathbf{H}\mathbf{H}^T) = 1$, as the spline basis vectors are not semi-orthogonal in general.





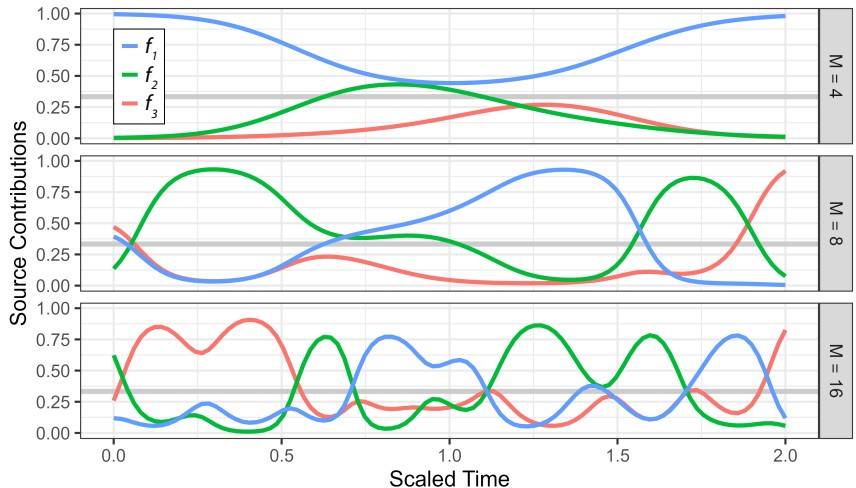

**Figure 2.** Examples of the spline prior for three source contributions ($f_1$, $f_2$, $f_3$ where $f_1 + f_2 + f_3 = 1$) transformed to the simplex with the CLR transformation using different degrees of freedom $M$ that can be used to control the covariance of source contributions at separate points in time.

## 2.4 Model comparison

### 2.4.1 Data simulation

No timeseries datasets with known $N_2O$ production and consumption pathway contributions are available, therefore simulated data must be used for thorough comparison of models. The four models were compared by simulating the data generating process multiple times, and then comparing the resulting posterior sample with fixed 'truth' input values. The time series of source contributions $f$ and pool fractions reacted $r$ used to simulate the data are denoted by $\mathbf{F}^* = [\mathbf{f}_1^* \cdots \mathbf{f}_N^*]$ and $\mathbf{r}^* = [\mathbf{r}_1^* \cdots \mathbf{r}_N^*]$ and the mixing equation with Rayleigh fractionation is used (Eq. 3). Measurement generation is then repeated $Q$ times by sampling the source isotopic signature $S^{(q)} \in \mathbb{R}^{d \times K}$ and fractionation factor $\epsilon^{(q)} \in \mathbb{R}^d$ from their respective priors, and then adding independent Gaussian measurement errors $E_t^{(q)} \sim \mathcal{N}_d(0, \eta^2)$ with noise variance $\eta^2 \in \mathbb{R}^d$ for $q = 1, ..., Q$ (Fischer, 2023):

$$X_t^{(q)} = \mathbf{S}^{(q)} \mathbf{f}_t^* + \epsilon^{(q)} \ln(r_t^*) + E_t^{(q)} \tag{19}$$

Auxiliary data for source isotopic signatures and fractionation factors are taken from Yu et al. (2020) (Table 1). These values correspond to the major $N_2O$ sources, nitrification ($S_1$) and bacterial denitrification ($S_2$). Priors are uniform for the sources $S_j \sim \mathrm{Uni}(b_j, \Delta_j), j = 1, 2$ and Gaussian for the fractionation factor with variance matched to the reported bounds $\epsilon \sim \mathcal{N}(c, \frac{\Delta_\epsilon^2}{2})$.

The simulations are done on fixed parameter sets that intend to be illustrative for six given cases that might occur in reality. The focus is on investigation of temporal patterns; other fractionation scenarios have been explored previously using





**Table 1.** Prior distribution parameters for N$_2$O source isotopic signatures and the fractionation factor for consumption used to simulate data sets for model testing (Yu et al., 2020). Ranges for sources indicate the full range of previous data used to construct the uniform distribution, whereas for reduction the range indicates the standard deviation of the Gaussian distribution.

| | $b_1 \pm \Delta_1$ | $b_2 \pm \Delta_2$ | $b_3 \pm \Delta_3$ | $c \pm \Delta_\epsilon$ |
|---|---|---|---|---|
| Source | Nitrification | Denitrification | Fungal denitrification | Reduction |
| Abbreviation | Ni | bD | fD | Red |
| $\delta^{15}\text{N}^{\text{bulk}}$ | -55.5±17.0 | -25.25±55.1 | -38.50±15.0 | -6.4±2.7 |
| $\delta^{15}\text{N}^{\text{SP}}$ | 35.35±6.7 | -1.9±11.2 | 33.55±12.7 | -5.55±1.5 |
| $\delta^{18}\text{O}$ | 23.5±6.0 | 20.0±6.6 | 38.45±7.3 | -15.4±5.8 |

a stationary model set up (Lewicka-Szczebak et al., 2020; Lewicki et al., 2022). The true parameter time series $\mathbf{f}_t^*$ and $r_t^*$ are shown in Figure 3 and are sampled at $N = 32$ equally spaced time steps. One additional general example (GenE) is used for simulation with properties being less extreme than for the other six, which may be more representative of average datasets that would be encountered in practice. For GenE, a Gaussian error with magnitude $\eta = 5$ is used to sample $N = 64$ measurements $X_1, ..., X_N$. The fixed parameter values and the simulated data is shown in Figure 3.

Bayesian parameter estimation is then tested on each generated data set $\mathbf{X}^{(q)} = [X_1^{(q)} \cdots X_N^{(q)}] \in \mathbb{R}^{d \times N}$ for $q = 1, ..., Q$ individually and a total of $S$ posterior samples of all parameters is produced each time. The posterior samples shall be denoted by $\mathbf{F}^{(q,s)} = [\mathbf{f}_1^{(q,s)} \cdots \mathbf{f}_N^{(q,s)}]$ and $\mathbf{r}^{(q,s)} = [r_1^{(q,s)} \cdots r_N^{(q,s)}]$ respectively for $s = 1, ..., S$.

All experiments were run on an Intel Core i9-10900K CPU. The reported run times in Section 3.1 are for a single sampling chain, and in Section 3.3 the reported times are the maximum of four simultaneously run chains.

### 2.4.2 Measuring quality of inference

Sampling from the posterior distribution does not give unique point estimates for the parameters involved, and multiple ways of computing final parameter estimates exist. Most commonly the posterior mean is used as point estimate, although using the median could, for example, be a useful strategy for posterior distributions that are highly dissimilar to a Gaussian distribution.

The accuracy of the estimation can be assessed by computing the distance between these pointwise estimates and the true value using root mean squared error (RMSE) and mean average error (MAE). Although it would be possible to compute the metrics at each point in time, they are averaged for simpler model comparison:

$$\text{RMSE}_k^{(q)} := \sqrt{\frac{1}{N} \sum_{t=1}^{N} (\hat{\mathbf{F}}_{kt}^{(q)} - \mathbf{F}_{kt}^*)^2} \tag{20}$$

$$\text{MAE}_k^{(q)} := \frac{1}{N} \sum_{t=1}^{N} |\hat{\mathbf{F}}_{kt}^{(q)} - \mathbf{F}_{kt}^*| \tag{21}$$

Computations for $\mathbf{r}^{(q,s)}$ are analogous.





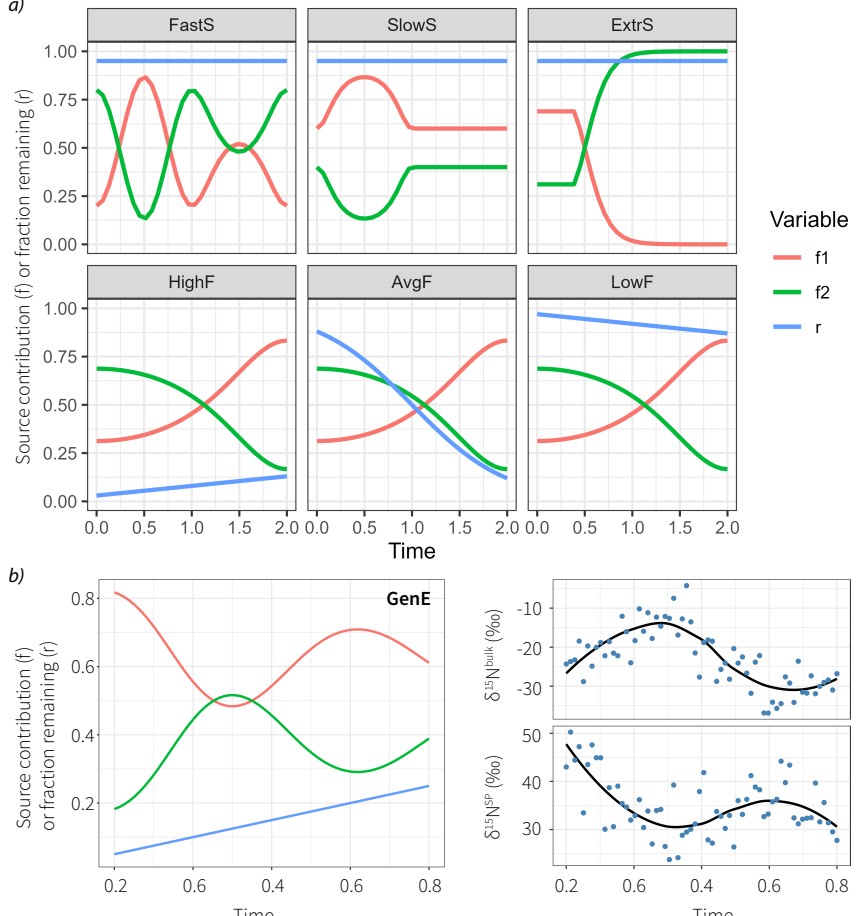

**Figure 3.** True parameter series used to simulate datasets with different properties. The production-consumption scenario corresponds to production by nitrification ($S_1$, red) and denitrification ($S_2$, green) following by mixing and subsequently reduction ($r$ = fraction remaining following reduction, blue) in complete denitrification. $a$) Six parameter series to test time series modelling, which illustrate: fast changing source contributions (FastS), slow changing source contributions (SlowS), extremal source contributions (ExtrS), high fractionation (HighF), average and variable fractionation (AvgF) and low fractionation (LowF). $b$) The general example (GenE) and the resultant isotopic timeseries, show measurement values simulated accordingly together with LOESS estimates in the right panel

Since the parameters to be estimated are interpreted as a time series, it makes sense to also compare specific time series information across the model estimates. The rate of change can be significantly confounded in the measurement time series, since the measurement errors follow a white noise distribution that introduces high frequency changes. The ability of models to filter this noise can be measured by comparing the rate of change which is approximated using first differences: $\Delta \mathbf{F}^*_{t,j} = \mathbf{F}^*_{t,j+1} - \mathbf{F}^*_{t,j}$ and $\Delta r^*_t = r^*_{t+1} - r^*_t$ for $t = 1, ..., N - 1$. The magnitude of changes is not necessarily relevant, since poor estimates of the magnitude of changes would also lead to poor pointwise comparison metrics. Therefore, the ratio of variances



of first differences shall serve as comparison metric for timeseries information, which can be understood as comparing a notion
of curvature or acceleration in the timeseries (Fischer, 2023):

$$\text{VarFD}_k^{(q)} = \frac{\sum_{t=1}^{N-1}\left(\Delta\mathbf{F}_{kt}^{(q)} - \sum_{j=1}^{N-1}\mathbf{F}_{kj}^{(q)}\right)^2}{\sum_{t=1}^{N-1}\left(\Delta\mathbf{F}_{kt}^{*} - \sum_{j=1}^{N-1}\mathbf{F}_{kj}^{*}\right)^2} \tag{22}$$

### 2.4.3 Metrics for comparison of Bayesian posterior distributions

Bayesian models are mainly used to derive pointwise estimates, but their advantage is the creation of a sample from the posterior
distribution. It is thus also important to take distribution properties into account. Posterior interval coverage is a useful metric
for to evaluate simulated data from the full Bayesian model, meaning in particular that $\mathbf{F}^*$ and $\mathbf{r}^*$ are sampled from their priors
as well. Since the data simulation for model comparison used here has fixed parameter values, interpreting interval coverage
becomes less meaningful. It is still practical to use the size of the credible interval as measure of uncertainty, and thus the
interval span can be compared, which is desired to be as small as possible given otherwise accurate estimates. The credible
interval with level $\gamma \in [0,1]$ is then the set excluding the tails with a proportion of $\frac{1-\gamma}{2}$ of the most extreme observations on
either side (Fischer, 2023):

$$I(\gamma) = \left[q(\frac{1-\gamma}{2}|\mathbf{X}), q(\frac{1+\gamma}{2}|\mathbf{X})\right] \tag{23}$$

Further metrics for the quality of the entire posterior distributions can be taken into consideration. Posterior predictive checks
are typically used in cases where no true values for the parameter estimates are available, in order to assess the models capability
of representing the input data well (Rubin, 1984). The posterior predictive distribution is the likelihood of hypothetical future
measurements calculated using the posterior distribution over parameter values in presence of the actually available data:

$$p(\tilde{\mathbf{X}}|\mathbf{X}) = \int p(\tilde{\mathbf{X}}|\mathbf{F},\mathbf{r})\pi(\mathbf{F},\mathbf{r}|\mathbf{X})d\mathbf{F}d\mathbf{r} \tag{24}$$

This posterior predictive distribution is not unique due to the auxiliary parameters $\mathbf{S}$ and $\epsilon$. It is unclear whether the posterior
predictive distribution should be proportional to the marginalized likelihood $p(\tilde{\mathbf{X}}|\mathbf{F},\mathbf{r})$ or rather the likelihood conditioned
on the auxiliary parameters $p(\tilde{\mathbf{X}}|\mathbf{F},\mathbf{r},\mathbf{S},\epsilon)$. This discrepancy renders comparison of predictive density values across dataset
simulations $\mathbf{X}^{(q)}$ ineffective, since the models fit a joint posterior and thus assume that future data must be sampled using
identical auxiliary parameter values, whereas the simulation resamples their values $\mathbf{S}^{(q)}$ and $\epsilon^{(q)}$ every time.

The log pointwise predictive density (LPPD) is a metric for the quality of the posterior predictive density, and thus by proxy
of the Bayesian model. It is computed by evaluating the predictive density at the original data points and can be approximated
using the posterior samples (Gelman et al., 2014):

$$\text{LPPD}^{(q)} = \ln \int p(\mathbf{X}|\mathbf{F},\mathbf{r})\pi(\mathbf{F},\mathbf{r}|\mathbf{X})d\mathbf{F}d\mathbf{r} \approx \frac{1}{S}\sum_{s=1}^{S} p(\mathbf{X}|\mathbf{F}^{(s)}) \tag{25}$$





## 3    Results and discussion

### 3.1    Sampling software

Bayesian models can be implemented using different sampling strategies. The most commonly used sampling libraries are `JAGS`, using Gibbs sampling, and `Stan`, using Hamiltonian Monte Carlo sampling. We compared two implementations in

three different settings:

- Using the original **stationary** FRAME estimator for one single time step given in Eq. 9,

- For the time series model using **independent** time steps described in Eq. 12,

- For the **hierarchical** Dirichlet-Gaussian process model described in Eq. 16, with joint estimation of the correlation length for source contributions and fractionation.

The general example GenE (Section 2.4.1) served as fixed truth value for $\mathbf{f}_t^*$ and $r_t^*$ to apply the models in the above mentioned three settings. The time series were sampled with $N = 64$ points, and for the stationary case the average over time is chosen as fixed value. Each implementation is run for $S = 10000$ sampling steps and the resulting effective sample size, total runtime in seconds and resulting effective samples per second are noted in Table 2. The effective sample sizes are computed using the calculation described by Kruschke (2014) and the reported number are averages over all parameters.

**Table 2.** Effective sample size, runtime in seconds and effective samples per second for the `Stan` and `JAGS` sampler over $S = 10000$ sampling steps for three different model set ups.

|  |  | $N_{eff}$ | Time (s) | $N_{eff}$/s |
|---|---|---|---|---|
| **Stationary** | Stan | 4 482 | 10 | 454 |
|  | JAGS | 5 131 | 0.5 | 9 162 |
| **Independent** | Stan | 6 350 | 66 | 97 |
|  | JAGS | 6 559 | 22 | 292 |
| **Hierarchical** | Stan | 11 538 | 1 477 | 8 |
|  | JAGS | 10 | 1527 | - |

`JAGS` outperforms `Stan` for the stationary and independent case by having more effective samples in the shorter amount of time. However, the hierarchical model could only be efficiently sampled by `Stan`, although it took a comparatively long time. These findings are consistent with results on linear models using different numbers of parameters (Beraha et al., 2021). `Stan` appears to be a good option for time series models which inherently have a lot of parameters. The cases where `JAGS` is faster have sufficiently short sampling times for both libraries, therefore `Stan` was used for all subsequent applications.




### 3.2 Prior distribution for the fraction remaining $r$

Sampling the prior distribution of the fractionation weight $r$ for closed system Rayleigh fractionation is challenging, because it is connected through the non-linear logarithm to the effect on measurements. Although a uniform prior usually does not inform the posterior about anything except the boundaries, the effect of the logarithm on the posterior is much more unclear. Different choices for prior distributions of $r$ were thus tested for their effect on the generated posterior sample. The simulated dataset used 17 different values for the fractionation index $d$ ranging from 0.05 to 0.95. Source contributions were fixed to $f_1^* = 0.7$ and $f_2^* = 0.3$. Each value of $r$ is used to generate $Q = 64$ data points $X^{(1)}, ..., X^{(Q)}$ with measurement error $\eta = 4$ for a total of 1088 data points (Figure 4$a$). The stationary inference model given in Eq. 9 is fitted to each point individually, which makes this setting analogous to the inference procedure used in the original FRAME model (Lewicki et al., 2022).

Using a uniform prior for the fraction reacted $\pi(r) \propto 1$ is the natural choice used as standard by all models. This was compared to the Jeffreys prior, which was derived to be $\pi(r) \propto \frac{1}{r}$ and thus is an improper prior (see Section 2.2 and Fischer (2023) for details). A middle ground between these two choices was given by the beta prior $r \sim B(\frac{1}{2}, 1)$, which has $\pi(r) \sim \frac{1}{\sqrt{r}}$. The first argument of the beta distribution can also be used as a free parameter $\alpha \in (0, 1]$ alongside a uniform hyperprior to construct the hierarchical model $r \sim B(\alpha, 1), \alpha \sim \text{Uni}(0, 1)$.

Each data point was supplied to the model individually and $S = 5000$ posterior samples were generated. The samples were combined for each value of $r$ to marginalize over the distributions of the auxiliary parameters (Figure 4$b$). The source $S_2$ (bD) has a wide prior distribution (Table 1) which confounds with the effect of fractionation and introduces high uncertainty. Since this uncertainty is proportional to the logarithm of $r$, the effect on the posterior distribution is much more pronounced when $r$ is high. The Jeffreys prior introduces a shift towards lower values compared to the uniform prior with the beta prior being in between the two. Inference performance was compared by taking the posterior means $\hat{\mathbf{f}}^{(q)} = \frac{1}{S} \sum_{s=1}^{S} \mathbf{f}^{(q,s)}$ and $\hat{\mathbf{r}}^{(q)} = \frac{1}{S} \sum_{s=1}^{S} \mathbf{r}^{(q,s)}$ and comparing them against the true values (Figure 4$c$). Clearly, the Jeffreys prior performs worst for source contributions $f$, with the uniform prior performing best, closely followed from the beta and hierarchical prior. Regarding the fraction remaining $r$, the Jeffreys prior performs best for low values of $r$, with the uniform prior performing worst, however this relationship switches at about $r = 0.4$ to the opposite. Therefore, choosing any prior can be justified if one expects the fractionation index to be in a certain range. The standard deviation between different repetitions however clearly shows that the effect of prior choice is overwhelmed by the variation introduced through the distribution of the sources, due to the large uncertainty in the source priors.

### 3.3 Model comparison on simulated timeseries

The four classes of models were compared for performance on timeseries data across a wide range of scenarios representative of edge cases that might occur in reality. For these experiments, fixed parameter values for source contributions $\mathbf{F}^*$ and fraction remaining $\mathbf{r}^*$ were sampled as described in Section 2.4.1. The model configurations to be compared are summarised in Table 3.



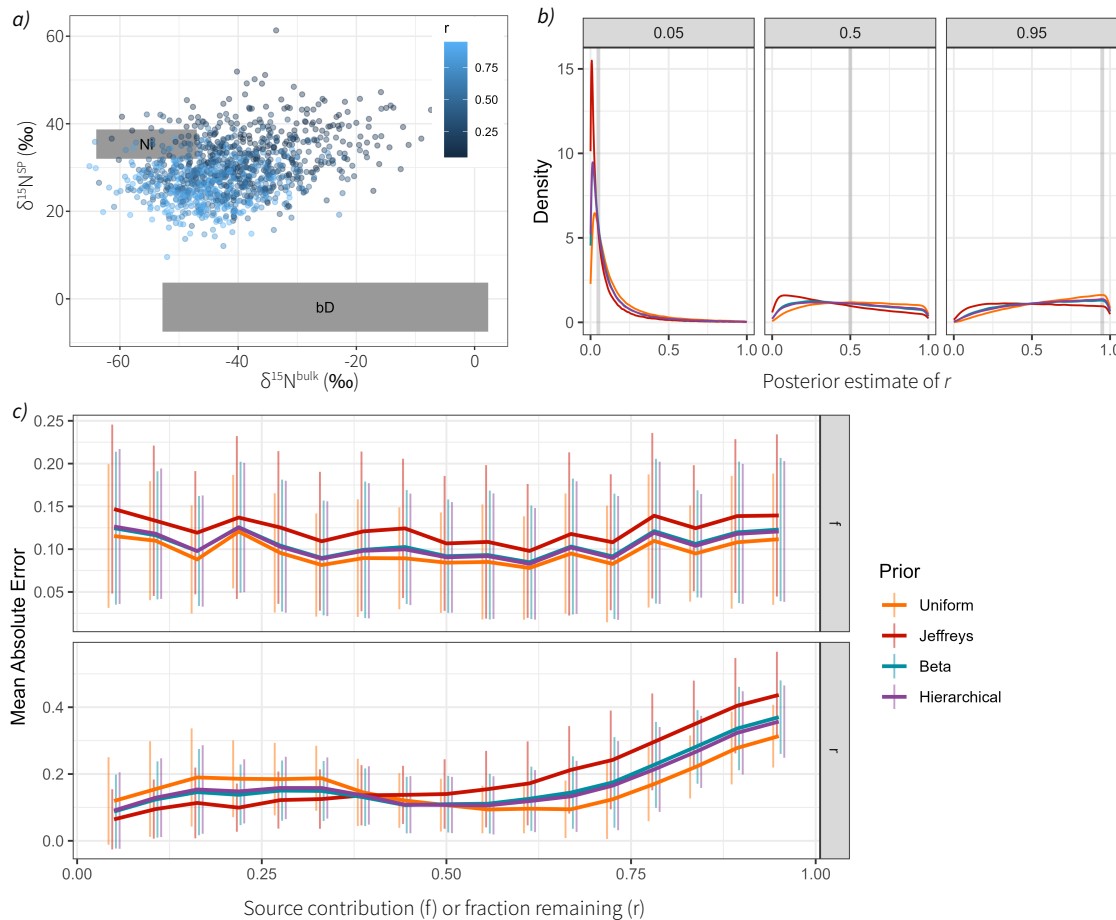

**Figure 4.** Comparison of different priors for the fraction remaining $r$. $a$) Dual isotope plot for the simulated data, constructed with two sources and different values of $r$. Data points are coloured by the 'true' value of $r$. $b$) Posterior densities of the fractionation weight $r$ averaged across simulations for 'true' $r$ values of 0.05, 0.5 and 0.95 using different prior distributions. $c$) Mean absolute error of Bayesian models using different prior distributions of $r$, shown for different 'true' values of $r$ and $f$. Vertical lines indicate the standard deviation over $Q = 64$ repetitions. The performance on source contributions is identical for both sources (Ni and bD), since they are perfectly correlated, so only one panel is shown for $f$.

### 3.3.1 Comparison of model performance

The examples described in subsection 2.4.1 were sampled for $Q = 64$ repetitions. Measurements were simulated with Gaussian measurement error of magnitude $\eta = 5$. The posteriors were sampled for a total of $S = 10000$ steps using four parallel chains. Goodness of estimation was quantified with estimates computed as the posterior means $\hat{\mathbf{f}}^{(q)} = \frac{1}{S} \sum_{s=1}^{S} \mathbf{f}^{(q,s)}$ and $\hat{\mathbf{r}}^{(q)} = \frac{1}{S} \sum_{s=1}^{S} \mathbf{r}^{(q,s)}$ and taking the mean absolute error (Eq. 21) to the ground truth (Figure 5$a$). Conclusions using the root mean squared error or with posterior medians were similar. All models using fixed hyperparameters use default values that are





**Table 3.** Model configurations used to compare performance on timeseries data across different edge cases, for the four model classes described in Section 2.3. 'Abbreviation' refer to the abbrevations for each configuration used in subsequent discussion.

| Model class | Abbreviation | Description |
|---|---|---|
| Independent | Independent | Independent time step model described in Eq. 12 |
| | Hierarchical | Hierarchical independent time step model with gamma hyperprior for concentration $\sigma$ described in Eq. 13 |
| Gaussian process prior | GP ($\rho = 1$) | Gaussian process prior on measurements with $\rho = 1$ described in Eq. 15 |
| | GP (latent) | Gaussian process prior on measurements with $\rho = 1$ using latent variable formulation described in Eq. 14 |
| | GP (hier.) | Gaussian process prior on measurements with inverse gamma hyperprior on $\rho$ |
| | GP (latent, hier.) | Gaussian process prior on measurements with with inverse gamma hyperprior on $\rho$ using latent variable formulation |
| Dirichlet-GP prior | DGP (CLR, $\rho = 1$) | DGP prior using CLR transform and $\rho = 1$, $\sigma = 1$ described in Eq. 16 |
| | DGP (ILR, $\rho = 1$) | DGP prior using ILR transform and $\rho = 1$, $\sigma = 1$ described in Equation Eq. 17 |
| | DGP (CLR, hier.) | DGP prior using CLR transform and $\sigma = 1$ with inverse gamma hyperprior on $\rho$ |
| | DGP (ILR, hier.) | DGP prior using ILR transform and $\sigma = 1$ with inverse gamma hyperprior on $\rho$ |
| Spline-based prior | Spline (CLR) | B-spline GLM using CLR link function having $M = 8$ for source contributions and $M = 4$ for fraction remaining described in Eq. 18 |
| | Spline (ILR) | B-spline GLM using ILR link function and $M = 8$ for source contributions and $M = 4$ for fraction remaining |
| | Spline (Laplace) | B-spline GLM using ILR link function and $M = 8$ for source contributions and $M = 4$ for fraction remaining with Laplace prior on coefficients |

not specifically tuned for the examples at hand. Therefore the reported performance is not indicative of best case performance and only shows the quality of the chosen values. Hierarchical models do not have this problem since they can estimate the

385 hyperparameters for each example specifically.

Overall performance was best for hierarchical DGP models and spline GLMs (Figure 5*a*). Estimation of fraction reacted $r$ is relatively poor in most cases (MAE $> 0.4$), however spline models perform well for all cases (MAE usually $< 0.3$), particularly for extremal fractionation amounts (all examples except AvgF). The default number of degrees of freedom that spline models use seems surprisingly robust in all examples, whereas the default correlation length of Gaussian processes does not. Gaussian

390 process priors on measurements appear to be slightly worse than DGP priors and spline-based priors, especially for examples SlowS and ExtrS. Independent time step models have worse performance than the rest for all examples and the hierarchical extension to it only has good performance in examples SlowS and ExtrS, which represent cases where concentration to sources is either very low or very high. In other cases using flat priors as default values seems to work best.

Spline GLMs have very small errors on fraction consumed $r$ whenever the value is slowly changing and close to 0 or 1.

395 This could suggest that the chosen hyperparameters are suitable for all examples. Another plausible explanation is the fact that



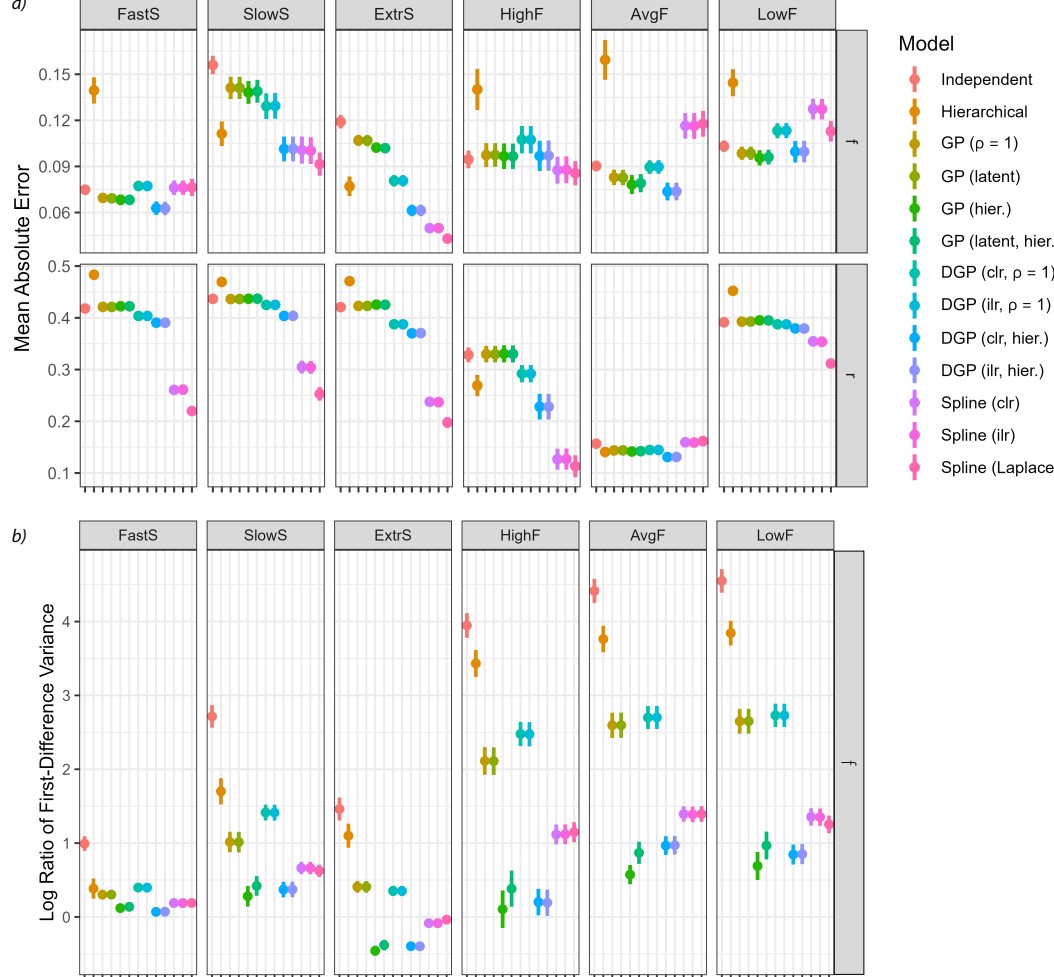

**Figure 5.** Model performance for different configurations (see Table 3 for details: fast changing source contributions (FastS), slow changing source contributions (SlowS), extremal source contributions (ExtrS), high fractionation (HighF), average and variable fractionation (AvgF) and low fractionation (LowF)). $a$) Mean absolute errors for all examples averaged over $Q = 64$ dataset simulations with standard deviations shown as vertical lines. The results for the two source contributions were identical since they are perfectly correlated, so the plots only show one result for $f$ in addition to the fraction remaining $r$. $b$) Log variance ratio of first differences of the estimated time series against the true values for source contributions $f$. The time series for fraction remaining $r$ are linear or constant in most examples and are thus not suitable to be used for model comparison with this metric.

the spline bases used have an intercept term, which allows the center of estimation to freely move, whereas DGP models do not. For the source contributions this likely does not matter, but allowing the Gaussian process to have non-zero mean could be beneficial for estimating $r$. An additional spline model was added with Laplace priors on the coefficients, which seems to be beneficial in cases where parameters are close to their boundaries since the prior allows for values farther from zero. The



model using Gaussian process priors on measurements was implemented using both formulations with latent variables and with analytically computed likelihood. Performance is identical in all cases, strongly indicating that the formulations are equivalent for parameter estimation. Additionally, using CLR or ILR transformations for DGP models and Spline GLMs does not make a difference in estimation accuracy, as is to be expected from their derivations.

Time series models are not only expected to give accurate estimates of mean source contributions and fractionation, but the resulting time series should also have similar properties to the ground truth. The variance ratio of first differences (Eq. 25) measures accuracy of the estimated curvature (Figure 5$b$). Clearly, the hierarchical Gaussian process and DGP models estimate the correlation length well, resulting in a time series with similar rates of change than the true values. Spline GLMs perform well, especially for the examples FastS, SlowS and ExtrS. Independent time step models result in high variation which is due to the fact that the measurement noise is not adequately filtered. The fixed correlation length Gaussian processes seem to have misspecified hyperparameters, since they also overestimate the rates of change in the time series.

Fitting times ranged from 20 seconds to 72 seconds on average for the $S = 10000$ posterior samples generated by each model, split into 2 500 over 4 chains (see Fischer (2023) for details). Hierarchical models tend to be slowest due to the additional parameters and repeated matrix decompositions that need to be computed, whereas fixed parameter models, especially independent time steps and spline GLMs, are sampled fastest. Spline models using the Laplace prior have long fitting times which could indicate that the high parameter values - which are allowed due to weaker regularization of of parameter ranges far from zero - are not sufficiently identifiable, resulting in slow sample generation.

### 3.3.2 Influence of fractionation extent

Models can have varying performance at different reaction extents and thus different levels of fractionation. For this reason, a timeseries of source contribution values $\mathbf{f}_t^*$ was taken and paired with different constant fractionation values $r_t^* = r^*$ to generate measurements and monitor performance. In total, 17 equally spaced fractionation values ranging from $r_* = 0.02$ to 0.98 were used and a total of $Q = 32$ datasets were generated per value. True values for source contributions were taken from the general example GenE (Section 2.4.1) and a measurement error magnitude of $\eta = 5$ was used. This experiment was done only with representative models of the four main model classes in order to reduce the number of comparisons. The model configurations used were the independent time step model, the Gaussian process prior on measurements with hierarchical estimation of correlation length, DGP prior with hierarchical estimation of correlation length, and spline-based GLM with fixed hyperparameters for degrees of freedom.

Overall performance is best when $r^*$ is close to 0.5 (Figure 6$a$). Estimation is also more accurate with very low values of $r^*$, because small $r^*$ values have large impacts on isotopic measurements and thus estimation can become more accurate. Spline models are expected to perform well here because the time series of fraction remaining is constant, which can be reflected in the low degrees of freedom used. The different model classes seem to be equally affected by changes in $r^*$ otherwise, showing that the choice of hyperparameters to reflect the situation of interest is more important than selecting a particular model.

Spans of 95% credible intervals can give additional insight into the pattern observed for the estimation accuracy over different values of $r^*$ (Figure 6$b$). If parameter estimation is good, then a smaller credible interval span shows a narrow posterior around





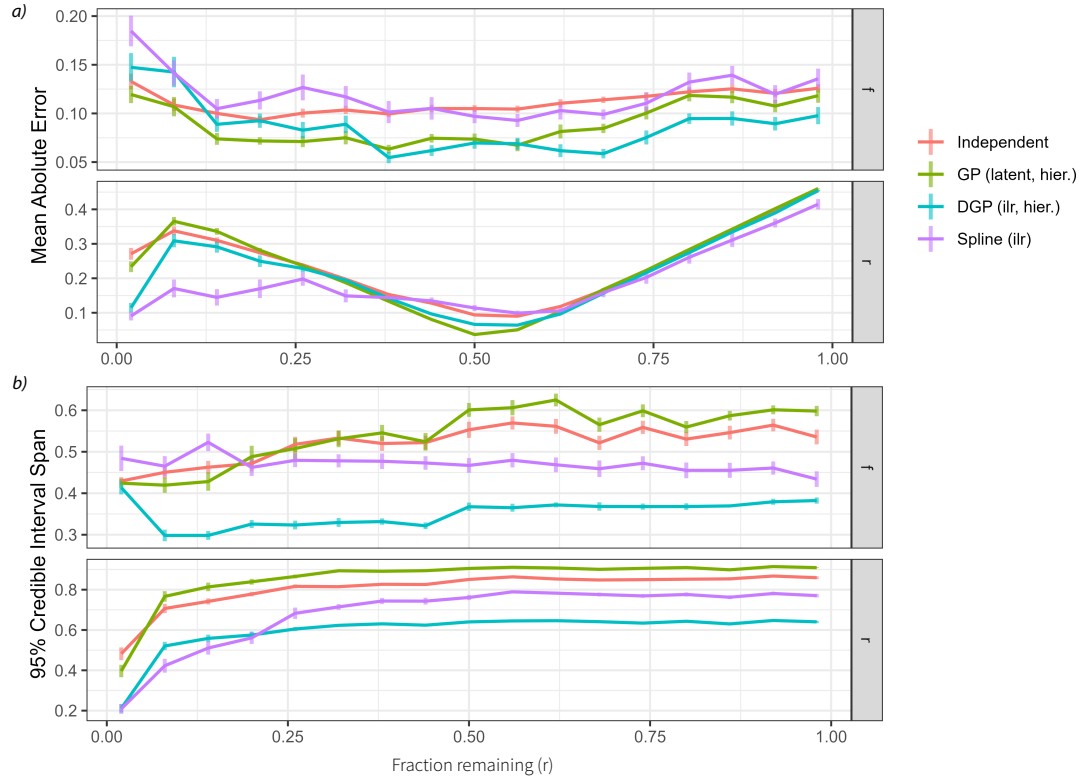

**Figure 6.** Impact of the fraction remaining $r$ on model performance. Each reported value is the average over $Q = 32$ dataset simulations with vertical lines indicating standard deviations. $a$) Mean absolute error of the four main model classes over different values of the fraction remaining $r$. $b$) 95% credible interval spans of the four main model classes over different values of the fraction remaining $r$.

the correct mean. DGP prior models have the smallest credible interval span for source contributions and fraction remaining. All other models have an interval width of over 0.75 for large values of fraction remaining, thus spanning over half of the possible domain. Clearly, due to the Rayleigh fractionation equation being non-linear in $r$, it is difficult to estimate large values of $r$ (resulting in low fractionation) with high accuracy. If the amount of remaining substrate is larger than 0.1, the data does not give sufficient information regarding $r$, so estimates of $r$ group around the prior mean of 0.5 and show large credible interval spans.

### 3.3.3 Influence of measurement noise

The main advantage that smooth models such as Gaussian processes and splines have over the independent time step assumption is that they promise to filter measurement noise and thus produce estimates that are more accurate and have a narrower posterior distribution. For this reason an experiment was conducted using values of source contributions $\mathbf{f}_t^*$ and fraction remaining $r_t^*$





from the general example GenE (Section 2.4.1) to simulate datasets with different levels of measurement noise. Noise values

range from $\eta = 0.5$ to $\eta = 20$, and for each separate value of $\eta$, a total of $Q = 32$ data sets were generated.

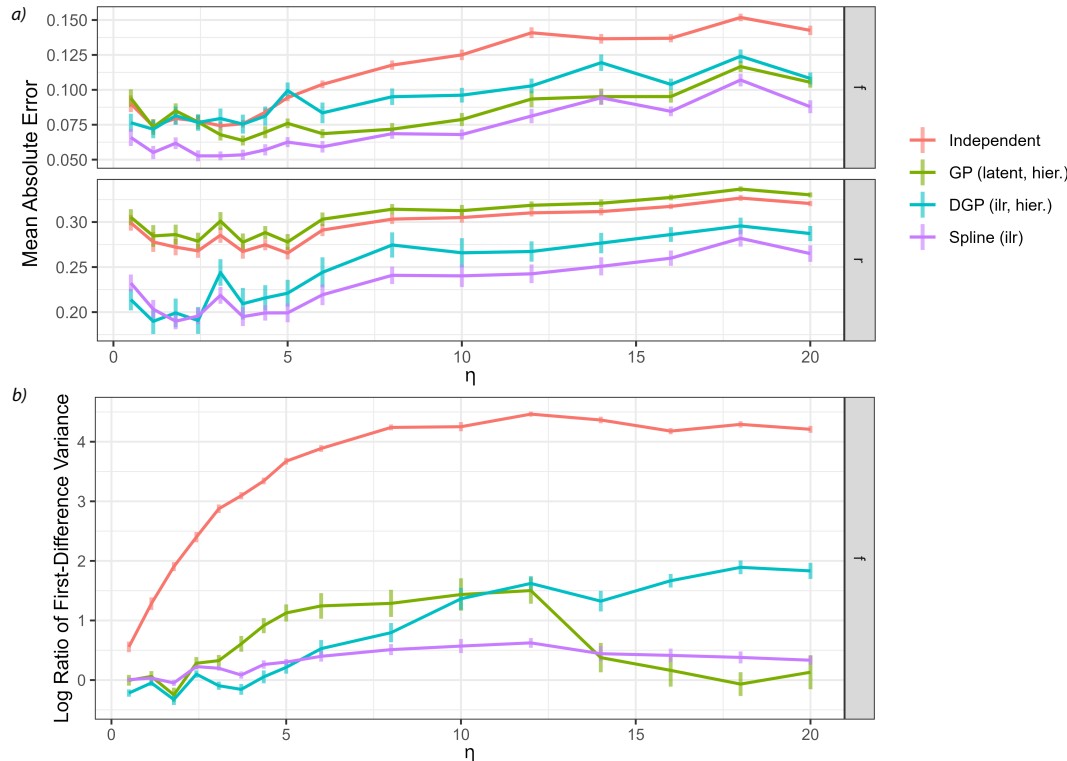

**Figure 7.** Impact of measurement noise on model performance. Each reported value is the average over $Q = 32$ dataset simulations with vertical lines indicating standard deviations. $a)$ Mean absolute error of estimation of the four main model classes over different measurement noise levels $\eta$. $b)$ Log variance ratio of first differences for the estimated time series against the true values for source contributions over different measurement noise levels $\eta$.

Performance of all models generally decreases with increased measurement noise as expected (Figure 7$a$). However, below $\eta = 5$, reduction in noise does not lead to further improvement in performance: Most of the estimation error at this noise level comes from the source endmember uncertainty rather than the measurement noise. The variance ratio of first differences (Figure 7$b$) can be used to assess the quality of the estimated timeseries in the presence of high frequency changes due to

measurement noise. Variance ratios of the independent time series model gradually increase with increasing measurement noise magnitude. All other models seem to filter the noise well, having much lower overestimations of the first difference variance. The hierarchical DGP model seems to be less equipped to deal with very high noise, which could simply be due to the fact that the weakly informative hyperprior on the correlation length is not suitable here. The spline GLM appears to have constant low values for the ratio of first difference variance, due to the fact that the fixed degrees of freedom predetermines the

smoothness of the estimates independent of measurement noise. Moreover, the model run time for the spline model was close




to the run time for the independent model, and less than half the run time for the GP and DGP models across all noise levels. These results show that use of the spline model can be particularly advantageous for data featuring high statistical uncertainty.

### 3.3.4 Impact of improvements in data quality

Estimation accuracy can be improved not only by choosing the right model but also by improving the quality of the data
available to the model. Several ways of adding more or higher quality data exist and the effects on model performance were studied in order to find what would be most beneficial. The above examples use two sources with two isotopic measurements, making the system well-determined. If additional isotopic measurements are available they can be added to make the system overdetermined and thus eliminate some noise. To investigate this approach, the additional isotopic measurement $\delta^{18}$O was added with source endmembers and uncertainty from Yu et al. (2020) (Table 1). The same dataset generation procedure as
in Section 3.3.1 was used with a total of $Q = 64$ datasets generated. Resulting improvement in estimation accuracy for the same two sources is shown in Figure 8$a$. It is worth noting that the additional measurement is not ideal in quality, with large uncertainty for the fractionation factor $\epsilon$. An improvement can clearly be seen for estimation of the fraction reacted $r$, especially in the examples HighF and AvgF. Very little improvement is seen for estimation of $f$. Spline GLMs improved the most, especially in their already good ability to estimate the fraction reacted.

Instead of adding additional measurements, efforts could be put into determining the endmembers of the sources and the fractionation factors more accurately, thus reducing uncertainty in the input data. To study this case an idealized set of sources and fractionation factors were selected to have the mixing and reduction line exactly perpendicular with an uncertainty of 10% in each dimension respective to the mean. This renders the mixing and fractionation components independent, since they cannot confound each other. We therefore set the endmembers for $S_1$ to $-1 \pm 0.2$ and $1 \pm 0.2$‰ for $\delta^{15}$N$^{\text{bulk}}$ and $\delta^{15}$N$^{\text{SP}}$
respectively, and for $S_2$ to $1 \pm 0.2$ and $-1 \pm 0.2$‰ for $\delta^{15}$N$^{\text{bulk}}$ and $\delta^{15}$N$^{\text{SP}}$ respectively. The fractionation factor was set to $1 \pm 0.1$‰ for both isotopes. Measurements sampled in this setting follow exactly the same procedure as in Section 3.3.1, but only use a Gaussian measurement error with magnitude $\eta = 0.1$. For each example $Q = 64$ datasets were generated and the mean absolute error of estimation is shown in Figure 8$b$.

Improving all uncertainties involved to a minimum has a great impact on model performance. Almost all mean absolute errors
of estimation are below an error margin of 0.05 for source contributions and below 0.1 for fraction remaining. Furthermore, model choice becomes less relevant, and even the independent time step models perform similarly to the other more sophisticated ones. Interestingly, the DGP model with fixed hyperparameters as well as the spline GLM with fixed spline basis underperform in source contribution estimation for example HighF. This could be evidence that the default parameters become less robust when noise is removed and they should be selected more carefully. Although this experiment is an extremely idealised case, it
shows the high potential for improvements in input data to enhance results, and moreover to make results more robust towards model configuration.





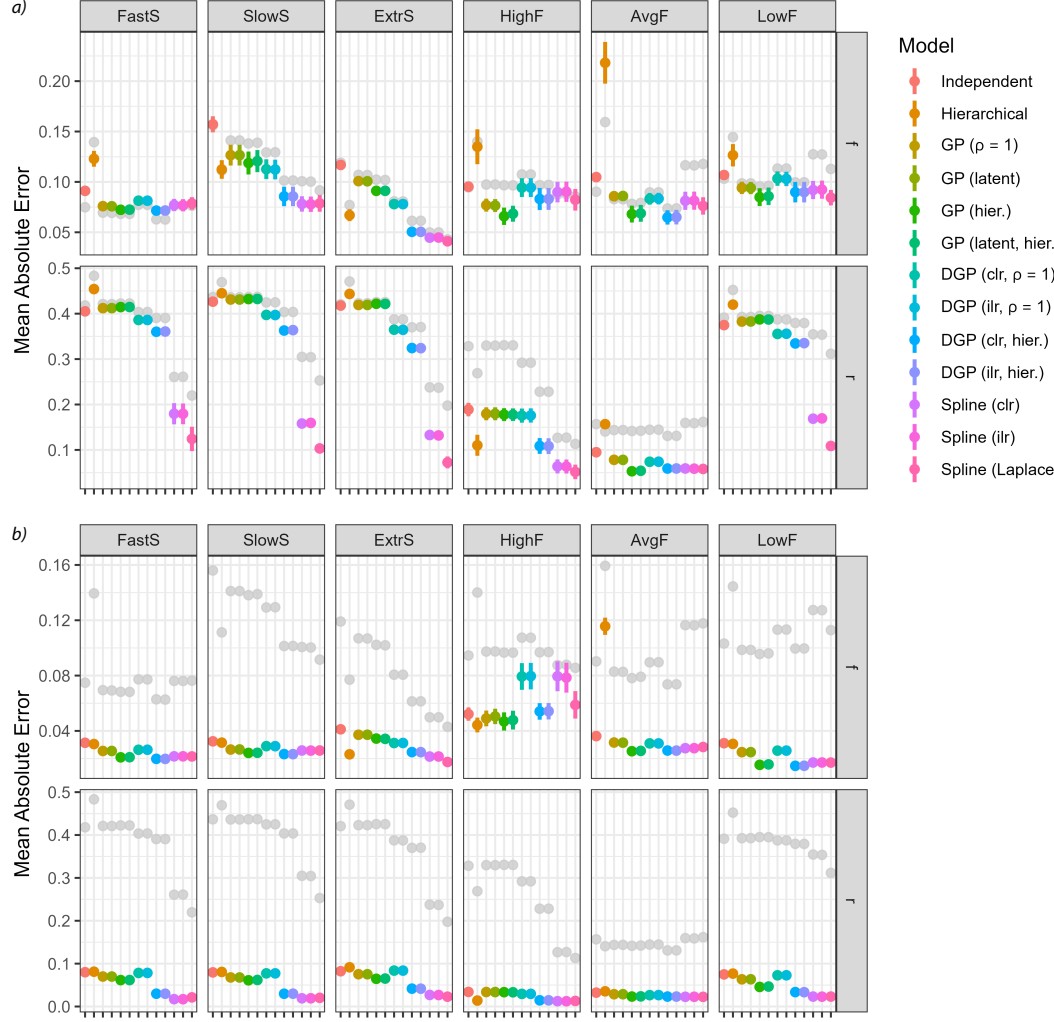

**Figure 8.** Model performance considering different improvements to the input data. The original model performance is shown in grey. *a)* Mean absolute error for all models on $Q = 64$ generated datasets using one additional isotopic measurement ($\delta^{18}$O). *b)* Mean absolute error for all models on $Q = 64$ generated datasets using the idealized sources and fractionation factor with 10% uncertainty in isotopic endmembers.

## 3.4 Application of the model to real and simulated datasets

In this section, we will present use of the TimeFRAME package for the analysis of simulated and real datasets, to illustrate aspects of model configuration choice and output data under different scenarios.





### 3.4.1 Application to the general simulated timeseries

The main purpose of the time series models is to provide estimates of source contribution and fractionation weights with uncertainty. In the sections above, only the performance metrics aggregated over many simulations have been shown. To illustrate the modeling capabilities the representative general example (GenE) was simulated from fixed parameter values and the inference results are shown in comparison to the true values.

In order to run the Bayesian models and estimate source contributions and fractionation over time, the auxiliary distributions of the source isotopic signatures $\mathbf{S}$ and the fractionation factor $\epsilon$ as well as the noise magnitude $\eta$ must be supplied in addition to the dataset. Three different model classes were run to illustrate the computed output: i) the independent time step model described in Eq. 12, ii) the spline GLM described in Eq. 18, and iii) the hierarchical DGP prior model described in Eq. 17. From the output that the models produce, either summary statistics of the posterior (such as its mean and quantiles) can be gathered, or the mean and credible intervals from all posterior time series sampled can be extracted as shown in Figure 9 .

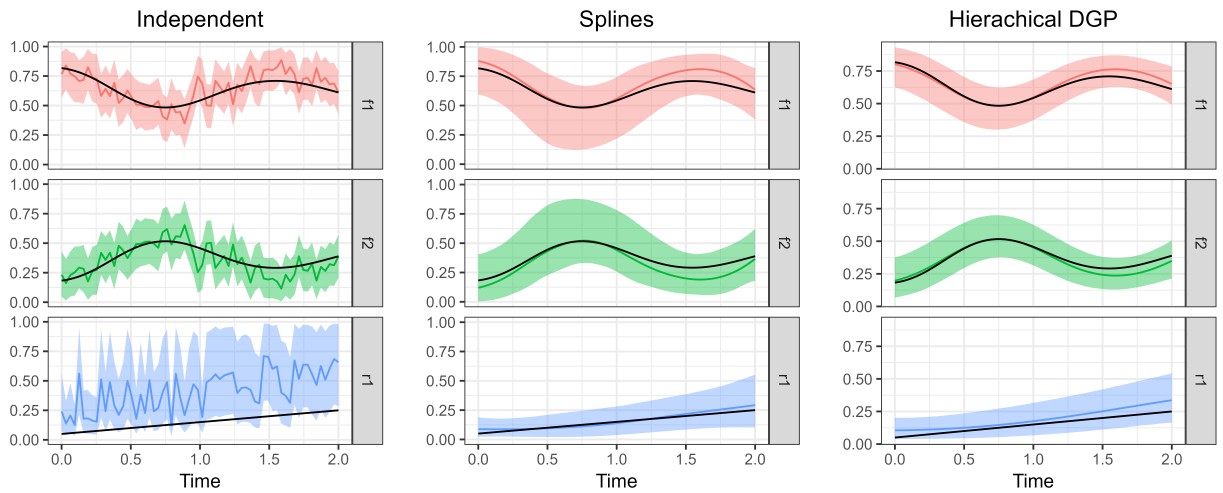

**Figure 9.** Posterior means of the three model types compared to the true parameter values. Shaded areas indicate 95% credible intervals and the true parameter values used to simulate the measurements are shown as black lines.

The independent time step model clearly shows poorer performance due to the large effect of measurement error on the estimated parameters. Nevertheless, the credible interval covers the true parameter values well and is reasonably narrow. Estimation of the fraction remaining $r$ seems to be biased toward higher values, which could be due to overlap with the variation in source isotopic signatures. The B-spline basis for the GLM seems to have default values for degrees of freedom that are fairly optimal in this case. The time series of parameter estimates is now similarly smooth to the actual parameter series. Estimation using the hierarchical DGP prior model gives the best results: The time series are adequately smooth and estimates are close to the true values with narrow credible intervals.



### 3.4.2 Comparison of TimeFRAME and a dual isotope mapping approach on a stationary dataset

The samples used to compare the TimeFRAME model with the traditional dual isotope mapping approach were taken from Kenyan livestock enclosures (Kiswahili 'bomas') at Kapiti Research Station and Wildlife Conservancy of the International Livestock Research Institute (ILRI) located in the semi-arid savanna region (1°35.8' W - 1°40.9' W, 36°6.4' E - 37°10.3' E). The different samples represent the isotopic composition of $N_2O$ taken from boma clusters of varying age classes (0-5 years after abandonment). At Kapiti, bomas are setup in clusters of 3-4, which are used for the duration of approximately one month before setting up a new cluster. The sampling campaign was conducted in October 2021 in order to understand the underlying mechanisms of the huge $N_2O$ emissions observed in these systems (Butterbach-Bahl et al., 2020), and the findings will be published in a separate paper (Fang et al., submitted). The dataset contains measurements of $\delta^{15}N^{bulk}$, $\delta^{15}N^{SP}$ and $\delta^{18}O$ of $N_2O$ as well as $\delta^{15}N$ of soil. Stable isotope analysis of these samples had been done using a IRMS as described in (Verhoeven et al., 2019) and (Gallarotti et al., 2021). A dual isotope mapping approach was used to extract the production pathways and reduction extent for this dataset based on $\delta^{15}N^{SP}$ and $\delta^{18}O$ using the scenarios of mixing following by reduction (MR) and reduction followed by mixing (RM), described in detail in Fang et al., (submitted) and Ho et al. (2023) .

As the boma dataset is a set of independent measurements, TimeFRAME was applied to the data using the independent time steps model. The time axis was replaced by numbering of points in the dataset. The standard deviation of all isotope measurements was set to 1‰ as the measurement uncertainty was not quantified, however this may be a low estimate given the many sources of uncertainty from measurement error to international scale calibration. TimeFRAME was applied in two configurations:

1. Using only $\delta^{15}N^{SP}$ and $\delta^{18}O$ and Ni and bD pathways, to closely mimic the configuration of the dual isotope MR approach, and

2. Using $\delta^{15}N^{bulk}$, $\delta^{15}N^{SP}$ and $\delta^{18}O$ with Ni, bD and fD (fungal denitrification) pathways, which is the most detailed configuration available with this data.

The agreement between the dual isotope method and the analogous TimeFRAME 2-isotope implementation is very good (mean absolute deviation of 8% and 15% for bD/Ni and reduction respectively, Figure 10). TimeFRAME offers the major advantage that uncertainty bounds for the prior are incorporated and thus calculated for the posterior. Moreover, using the TimeFRAME package functions, the calculations are reproducible, can be performed in just two lines of code, and can be easily adapted to consider different endmembers and model set ups. The TimeFRAME 3-isotope implementation shows very different results to the other estimates due to the inclusion of $\delta^{15}N^{bulk}$ and the fungal denitrification pathway. This pathway has high $\delta^{15}N^{SP}$ (Table 1) and thus strongly impacts the model estimates of nitrification and reduction, which are also evidenced by high $\delta^{15}N^{SP}$.

The pathway estimates were used to reconstruct the isotope measurements, and the RMSE between true measurements and reconstructed measurements was found as an estimate of model performance in the absence of true knowledge of pathways (Table 4). The TimeFRAME 2-isotope implementation is able to reproduce the isotopic data more accurately than the dual





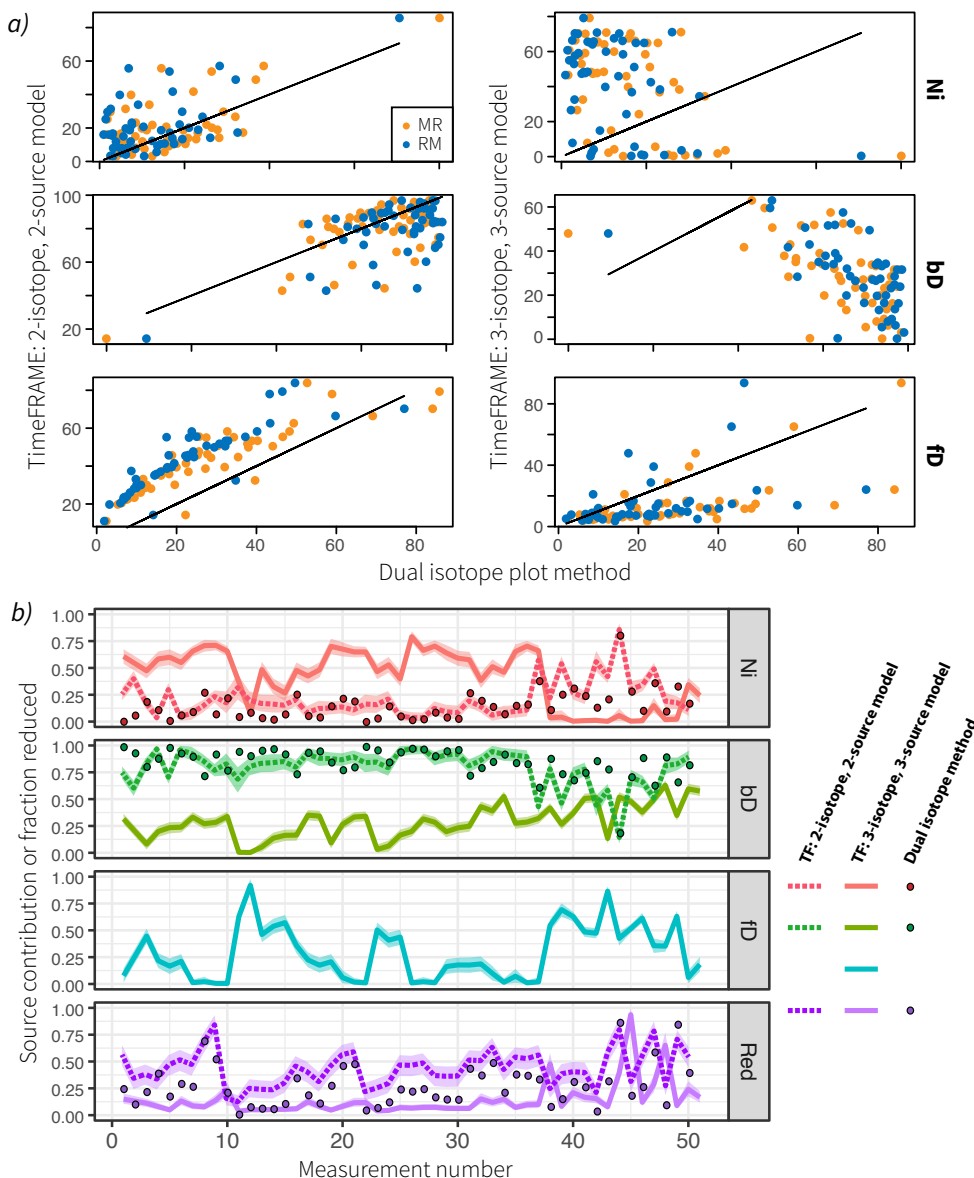

**Figure 10.** Comparison of TimeFRAME results with pathways estimates from a dual isotope plot approach. TimeFRAME was applied in two configurations, i) using only $\delta^{15}N^{SP}$ and $\delta^{18}O$ and Ni and bD pathways, and ii) using $\delta^{15}N^{bulk}$, $\delta^{15}N^{SP}$ and $\delta^{18}O$ with Ni, bD and fD pathways. The dual isotope approach used $\delta^{15}N^{SP}$ and $\delta^{18}O$ and Ni and bD pathways, in configurations MR (mixing then reduction) and RM (reduction then mixing). *a*) A 1:1 comparison of estimates from the two TimeFRAME configurations and the dual isotope plot MR and RM methods. *b*) A plot of the estimates for each pathway from the two TimeFRAME configurations and from the dual isotope MR method.

isotope plot due to the Bayesian optimization of the fit. The TimeFRAME 3-isotope implementation shows poorer RMSE due to the additional challenge of fitting $\delta^{15}N^{bulk}$ as well as the fD pathway. The difference between MR and RM implementations



**Table 4.** RMSE (‰) for isotopic data estimated using pathway contributions from the two TimeFRAME and the two dual isotope model configurations, compared to the measured isotopic data used as input for the models.

|  | $\delta^{15}N^{bulk}$ | $\delta^{15}N^{SP}$ | $\delta^{18}O$ |
|---|---|---|---|
| TimeFRAME, 2-iso., 2 source | - | 6 | 19 |
| TimeFRAME, 3-iso., 3 source | 36 | 22 | 17 |
| Dual isotope, MR | - | 7 | 23 |
| Dual isotope, RM | - | 7 | 23 |

of the dual isotope approach is minimal, showing that model configuration and uncertainty in endmembers is far more important for results than the specific formulation of the fractionation and mixing equation.

These results show the importance of considering different pathways and model configurations. TimeFRAME users should aim to include as much isotopic data as possible, and to use other complementary approaches such as microbial ecology to constrain potential production and consumption pathways; for example, to decide whether it is appropriate to include fungal dentrification. Users should consider both the estimated uncertainty for a particular model set up, provided by the TimeFRAME package, as well as the variation between estimates for different plausible scenarios.

**3.4.3    Comparison of timeseries analysis with existing approaches**

TimeFRAME was applied to two irregularly time-spaced datasets from soil incubations at different soil moisture levels (L1 = drier = 55-66% WFPS; L2 = wetter = 69-82% WFPS). The L1 and L2 incubations were sampled on eight and eleven dates respectively with between one and seven duplicate measurements taken on each sampling date, and a total of 41 and 24 measurements made respectively. The incubations are described in detail in Lewicka-Szczebak et al. (2020). TimeFRAME

was compared to results from the 3DIM/FRAME model (Lewicka-Szczebak et al., 2020; Lewicki et al., 2022), with both models considering four pathways (bacterial denitrification, nitrifier denitrification, fungal denitrification, nitrification) as well as $N_2O$ reduction using the endmembers and fractionation factors reported in Lewicka-Szczebak et al. (2020). FRAME solves the isotopic equations independently for each sampling date, whereas TimeFRAME spline and DGP implementations are able to consider temporal correlations between sampling times. Additionally, dual isotope mapping and [15]N labelling approaches

were compared, as described in Lewicka-Szczebak et al. (2020).

The agreement was good between pathway estimates from TimeFRAME and FRAME, although the spline implementation estimated lower reduction than other methods (Figure 11). Agreement with the mapping approach was very poor for bD and good for reduction, reflecting the low ability of the mapping approach to unravel pathway contributions with similar endmembers. Agreement with the [15]N gas flux method was good for reduction and acceptable for bD, considering the

denitrification contribution being quantified is not identical. The results clearly showed the influence of WFPS on bD and reduction, with both pathways increasing by 2% for every 1% increase in WFPS (Figure 11).





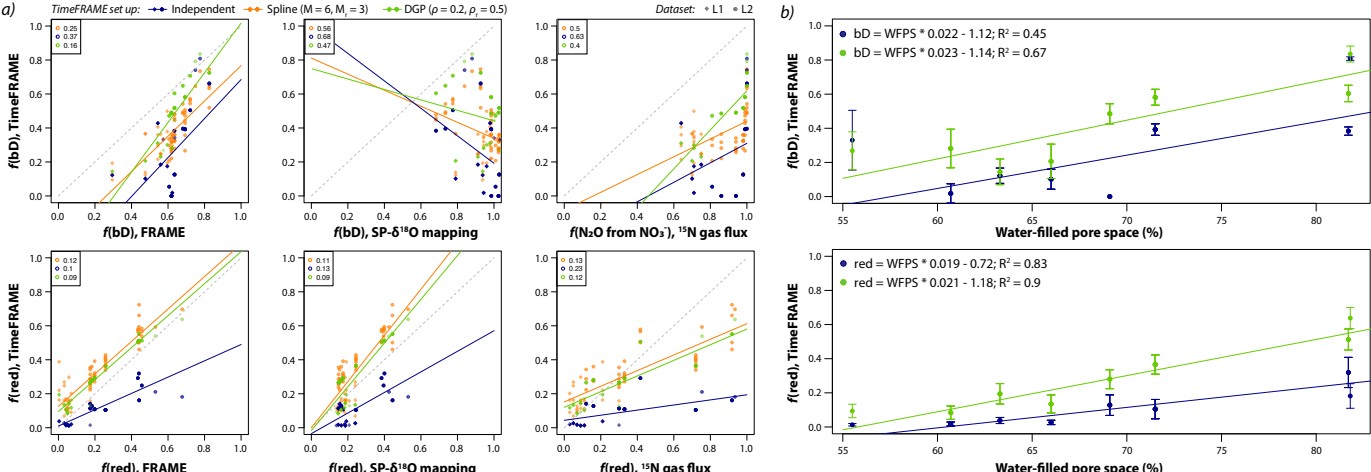

**Figure 11.** Using TimeFRAME to understand the impact of WFPS on N$_2$O production and consumption. *a*) Comparison of TimeFRAME pathway estimates (independent, spline (M = 6 and M$_r$ = 3) and DGP ($\rho$ = 0.2 and $\rho_r$ = 0.5) implementations) with estimates from the FRAME model, from an SP-$\delta^{18}$O mapping approach, and from a $^{15}$N-labelled gas flux approach. Comparisons are only shown for bD (bacterial denitrification) and reduction as other pathways are not estimate by the mapping and gas flux approaches. The gas flux approach does not strictly estimate bD, but the proportion of N$_2$O arising from NO$_3^-$ substrate. The lines show the linear regression and the legends show the mean absolute deviation for each comparison. *b*) Impact of WFPS on bD and reduction estimated using TimeFRAME spline and DGP results. The error bars show the estimated standard deviation at each point from the TimeFRAME fit. The legend shows details of the weighted linear regression (R lm() function weighted by $\frac{1}{\sigma}$) for each dataset.

## 4 Conclusions

The TimeFRAME data analysis package uses Bayesian hierarchical modelling to estimate production, mixing and consumption pathways based on isotopic measurements. The package was particularly developed for the analysis of N$_2$O isotopic data and

contains default isotopic endmembers and fractionation factors for N$_2$O, but the flexible implementation means it can also be applied to other trace gases such as CH$_4$ and CO$_2$. TimeFRAME provides a simple and standardised method for analysis of pathways based on isotopic datasets, which has previously been lacking. The package will contribute strongly to reproducibility and uncertainty quantification in the analysis of these datasets.

TimeFRAME has four main classes of model which have been extensively tested in a range of scenarios. For timeseries data,

the Dirichlet-Gaussian process and spline GLM priors show very good results. These models are able to smooth timeseries to reduce the impact of noisy data, and deliver good pathway quantification compared to the ground truth for simulated datasets. The independent timestep model is strongly impacted by measurement noise, but delivers good performance compared to the dual isotope mapping approach, with simpler and more reproducible implementation as well as uncertainty quantification.

Model application and testing showed that uncertainty in endmembers and fractionation factors was the major source of

uncertainty in pathway quantification. Reduction of uncertainty in these parameters will strongly improve the insights that can



be gained from isotopic data. Model set up is also critical: The sources/pathways chosen in the model strongly affect results, and should be informed based on any other relevant sources of information, for example profiling of the microbial community present at a measurement site. TimeFRAME provides a robust and powerful analysis tool but the accuracy of results gained from TimeFRAME depend on careful definition of the model set up and configuration by the user.

*Code and data availability.*

The TimeFRAME code and application data shown in the manuscript can be accessed at:

– https://gitlab.renkulab.io/eliza.harris/timeframe

The TimeFRAME package for direct installation with devtools is located at:

– https://github.com/elizaharris/TimeFRAME

The TimeFRAME user interface (Shiny app) is useful for first interactions with the mode. The TimeFRAME shiny app can be directly started:

– https://renkulab.io/projects/fischphi/n2o-pathway-analysis/sessions/new?autostart=1

Alternatively, session settings for the Renku platform deployment can be chosen before the app is initialised:

– https://renkulab.io/projects/fischphi/n2o-pathway-analysis/sessions/new

The development version of TimeFRAME, including the different edge scenarios explored in this manuscript, can be accessed at:

– https://gitlab.renkulab.io/fischphi/n2o-pathway-analysis

In particular, code used for the experiments can be found:

– https://gitlab.renkulab.io/fischphi/n2o-pathway-analysis/-/tree/main/experiments

*Author contributions.* EH and PF wrote the paper with comments and input from other coauthors. PF developed the TimeFRAME package and carried out extensive testing of the tool during his MSc thesis at ETH Zürich under the supervision of EH, FPC, ML and DLS. ML and
DLS developed the FRAME package and provided guidance for the expansion to TimeFRAME.

*Competing interests.* We declare no competing interests.

*Acknowledgements.* We thank Matti Barthel (USYS, ETHZ) for contribution of boma data. We acknowledge the support of the Swiss National Science Foundation in funding the 'N2O-SSA' project (200021_207348). DLS was supported by the Program 'Polish Returns' of the Polish National Agency of Academic Exchange.



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
