# Peer review of "Technical Note: TimeFRAME - A Bayesian Mixing Model to Unravel Isotopic Data and Quantify Trace Gas Production and Consumption Pathways for Timeseries Data"

_EGUsphere, 2023_

## Referee Comment (RC1)

This manuscript by Harris et al. documents and aims to validate a useful software package called "TimeFRAME". TimeFRAME can be used to analyse production, mixing and loss of trace gas compounds using isotope composition time series. It builds on the existing "FRAME" with 4 Bayesian hierarchical models. TimeFRAME provides a relevant and useful package to the scientific community, within the scope of biogeosciences. Although I believe the package can be useful, I see a major issue with the validation strategy of TimeFRAME as a discuss below. This needs to be addressed before considering this for publication. Additionally, I currently do not posses the expertise to fully evaluate the correctness of section 2.3. The manuscripts, written in clear language, is a bit long and at times somewhat unstructured, for which I have some remarks below as well.

A major issue I had after reading the manuscript is the applicability of TimeFRAME due to the assumption made to arrive at equation 3 on line 45. The authors acknowledge already that in this equation it is assumed that mixing and fractionation are separable. In addition, they acknowledge that this is an unlikely scenario, and it would very much depend on the use case if it is applicable. It is well known, that for most applications, when one or more loss terms is involved, accurate source apportionment becomes complicated (see for example Kaiser et al., 2006; Röckmann et al., 2011) and usually requires time resolved 2D or 3D modelling. Yet, the authors conclude that consumption pathways can be estimated using TimeFRAME. For this final generalized conclusion no proof is presented in this manuscript. In section 2.4.1 the data simulation experiments are explained. The data is generated by using equation 3. I.e. the forward model for data already has mixing and the loss process separated. It is thus expected that, given small uncertainty you retrieve back the inputs. So consumption pathways can be estimated only for the case where mixing and loss processes can be separated. For most applications, which the authors acknowledge (and there I fully agree), this is unlikely. So the experiment does

not support the general conclusion that TimeFRAME can quantify mixing and loss in general. I'm sceptical that it can, as soon as the leading assumption is violated. Moreover, this significantly reduces the impact of this package for the cases where it is validated. Then I wonder, with the current validation: to what extend TimeFRAME is different from the other packages mentioned by the authors (MixSIAR, simmr)?

Why did the authors not use a model that does consider mixing and loss at the same time to use for data generation? With a limited number of parameters, a system of differential equations can be designed to trace the relevant parameters. Then, you can see to what extend the linearised model used in TimeFRAME is capable of retrieving the relevant parameters in such a case. This would fully validate and allow the authors to quantify the circumstances under which the model is no longer applicable. Making the manuscript far more valuable. Moreover, it would present users with a methodology to come up with their own validation strategies when applying TimeFRAME to their own data.

General comments

Equations 1 and 2 (and therefore also 3 but equation 3 is discussed separately above) are approximations. See for example Chapter 4 Mook, 2000. Mook, 2000 noted that for equation 1 as presented in the manuscript by Harris et al., the induced error is indeed small. Nonetheless, as this paper is the fundamental reference for "TimeFRAME" this should at least be mentioned. Additionally, use "$\approx$" for equations where approximations are made to signal the user that the relation is not exact.

As mentioned I'm not fully equipped to judge the validity and correctness of section 2.3. Though I certainly see the relevance of documenting the techniques used in section 2.3, I don't understand it's purpose in the current text. I would be more interested to see a paragraph describing the strength and weaknesses of each of method. This can then be reflected on in the discussion. The technical description can be put in an appendix.

Results and Discussion section is long. The results are sometimes difficult to find between discussion paragraphs and also some parts that belong in the methodology section. Would it not be better to separate results and discussion? Some more suggestions:

1. Section 3.1 is not a result, it is a software design choice. A single paragraph in Section 2 could convey the same information: "We have settled on Stan over JAGS because of performance." Also, another column in table 3 could be used to specify runtime per experiment the give the author an idea of how long each computation takes. Then Table 2 can be removed altogether. (note I also have a question in the minor comments below on this topic)

2. Section 3.2 starts of with the need to test the prior distribution for the fraction remaining $r$. Shouldn't that be part of the methodology?

3. Table 3 is part of the methods. I suggest introducing the application of the different models in a separate section after 2.4.1. This section could alternatively benefit from a clear presentation of the different settings for the parameters in the model. (in other words what arguments are passed on to the function in the software package).

4. A paragraph introducing the different subsections in the results section would be very helpful to guide the reader, i.e. prior to section 3.1.

I miss a paragraph discussing the application of TimeFRAME to other trace gas isotope timeseries. They are mentioned in the introduction and abstract ($CH_4$, $CO_2$), but not discussed, this also clearly relates to my major issue. What can the authors see about applications to other isotopes, for example atmospheric $CH_4$, which has several loss terms in the Stratosphere, with temperature dependent isotope effects? The abstract seems to suggest that this software package is capable of dealing with that.

Although the technical description is certainly relevant, I miss a bit of an explanation how the user would determine the prior. For example, in Figure 2, if the user were to know a priori that at $f_1(t = 0) = 1$ how is that achieved? When should I use a Gaussian process prior and when a Dirichlet-Gaussian prior? It would be helpful to potential users, what kind of information needs to be defined per model. Throughout the text (For example L 457) recommendations on the usage of the different models can be found. It would be helpful to have a single paragraph (table?) that summarize the recommendations for different applications.

Some minor details caught my attention:

L10 here it states "..production, mixing and consumption..", however the rest of the text considers only 2 endmember mixing with a single consumption pathway. What if there are three? or what if there are two loss processes? Please update the abstract, title, and manuscript to be in line with what is presented.

L18 Consider updating this after addressing my comment on L468.- I think an additional isotope can be very useful in cases where endmembers are significantly different.

L 39 "... substrate being consumed before consumption..." reads a bit odd, simply put "..the substrate before its consumption.."

L 44 I could not find and access Fischer 2023. So the derivation of this is missing.

L 62 MixSIAR and simmr are mentioned here, but are not reflected on in the discussion. Would it not be informative to include a comparison without a sink? Or at least reflect on those packages? For mixing-only application is there a substantial difference? Does TimeFRAME offer a unique functionality?

L 79 The remainder of the text suggest that only one fractionation factor can be used?

L 219 Please refer to the original text, as I'm not sure if Fischer 2023 is appropriate here. (potentially other places as well)

L 221 Is it necessary for this text to fully explain the DGP process. Would it not be sufficient to simply state that a Dirichlet-Gaussian process is modelled?

L 272 Why not test with three sources. How well would your model distinguish the b1 and b2 sources for oxygen-18? Also, fractionation factors are often variable, for example with a temperature dependence. There is no discussion on the applicability of this model to that specific case.

Figure 3. You have only two sources whereas Table 1 lists three. Why is there no test case with all three sources? What can you conclude about more sources?

L 304 "... is a useful metric for to evaluate ..." is bad sentence, please rephrase

L 316 and elsewhere, it is recommended that mathematical operators should be written upright Roman. See (Cohen et al., 2007) chapter 4. So the differential $d$ in the integral should be d.

L 329 The conclusion to use Stan is based on timing tests as presented in Table 2. Yet there seems to be a fundamental difference in the sampling strategies. I assume that those don't matter for your application?

L 347 Doesn't this simply reflect the fact that, for large fractions remaining the effect on isotope measurements is simply so small that it is difficult to see? Especially in the light of the noted uncertainties in Table 1. In other words, if the fractionation constants are well know, and much larger, the change in isotope measurements is much larger and you would be able to see that. Therefore, I think it is strange to attribute this to the nature of the logarithm. In other words, you simply can't resolve $r$ very well given the values in Table 1.

L 370 what about uncertainty in the fractionation factors?

L 420 To what extent does the described experiment here relate to the experiment described in section 3.2

L 427 This is what I meant with my comment on L 347... Those two descriptions are in conflict. This seems to be the valid analysis.

L 468 "Very little improvement is seen for the estimation of $f$." I think this is to be expected. Looking at table 1, $\delta(^{18}O)$ source signatures are very similar for

nitrification and denitrification with in there respective uncertainties. I think you would see improvements when adding the third source, fungal denitrification. Why didn't the authors consider that for this experiment?

L 470 Is this in any case realistic? Natural variability can put a limit on end-member uncertainty reduction through measurements.

**References**

Cohen, E., T. Cvitas, J. Frey, B. Holmstrom, K. Kuchitsu, R. Marquardt, I. Mills, F. Pavese, M. Quack, J. Stohner, H. Strauss, M. Takami, and A. Thor (2007). *Quantities, Units, and Symbols in Physical Chemistry*. RSC Publishing. DOI: `10.1039/9781847557889`.

Kaiser, J., A. Engel, R. Borchers, and T. Röckmann (2006). "Probing stratospheric transport and chemistry with new balloon and aircraft observations of the meridional and vertical $N_2O$ isotope distribution". In: *Atmospheric Chemistry and Physics* 6.11, pp. 3535–3556. DOI: `10.5194/acp-6-3535-2006`.

Mook, W. G. (2000). *Environmental isotopes in the hydrological cycle*. Vol. 1. IHP-V, Technical Documents in Hydrology. UNESCO.

Röckmann, T., M. Brass, R. Borchers, and A. Engel (2011). "The isotopic composition of methane in the stratosphere: high-altitude balloon sample measurements". In: *Atmospheric Chemistry and Physics* 11.24, pp. 13287–13304. DOI: `10.5194/acp-11-13287-2011`.

---

## Author Comment (AC1)

**Reviewer 2: Response**

**E. Harris et al.**

We thank the reviewer for their positive comments regarding this manuscript. We respond to their comment:

*My only remark is that the methods are purely statistical approaches. Timeseries data can also be tackled with dynamic, mechanistic approaches (based on differential equations), which typically have few parameters to be fitted and can ingest more diverse data sets, i.e. including concentrations. While I am not claiming that the authors should also discuss those methods at length, it would be desirable that they at least mention this -alternative- approach to stable isotopic data analysis for timeseries data.*

We agree that this is an alternative approach to this problem that should be mentioned and we have added at L74:

Timeseries information can be added to isotopic models through statistical approaches using smoothing and other techniques to account for temporal autocorrelation and measurement noise, or through the application of dynamic approaches incorporating differential equations [1]. In TimeFRAME, we use the statistical approach as a natural extension to the implementation of FRAME; investigation of dynamical approaches may be challenging due to high uncertainties in all inputs and should be a focus of further research.

**References**

[1] W. Bonnaffé, B. C. Sheldon, and T. Coulson. Neural ordinary differential equations for ecological and evolutionary time-series analysis. *Methods in Ecology and Evolution*, 12(7):1301–1315, 2021.

---

## Author Comment (AC2)

**Reviewer 1: Response**

**E. Harris et al.**

We thank the reviewer for their detailed comments on this manuscript. We have responded to all comments, which has greatly improved the clarity of the manuscript. The comments are addressed below in the order in which they appear in the review. Line numbers refer to the pre-review manuscript.

1. *A major issue I had after reading the manuscript is the applicability of TimeFRAME due to the assumption made to arrive at equation 3 on line 45. The authors acknowledge already that in this equation it is assumed that mixing and fractionation are separable. In addition, they acknowledge that this is an unlikely scenario, and it would very much depend on the use case if it is applicable. It is well known, that for most applications, when one or more loss terms is involved, accurate source apportionment becomes complicated (see for example Kaiser et al., 2006; Röckmann et al., 2011) and usually requires time resolved 2D or 3D modelling. Yet, the authors conclude that consumption pathways can be estimated using TimeFRAME. For this final generalised conclusion no proof is presented in this manuscript. In section 2.4.1 the data simulation experiments are explained. The data is generated by using equation 3. I.e. the forward model for data already has mixing and the loss process separated. It is thus expected that, given small uncertainty you retrieve back the inputs. So consumption pathways can be estimated only for the case where mixing and loss processes can be separated. For most applications, which the authors acknowledge (and there I fully agree), this is unlikely. So the experiment does not support the general conclusion that TimeFRAME can quantify mixing and loss in general. I'm sceptical that it can, as soon as the leading assumption is violated. Moreover, this significantly reduces the impact of this package for the cases where it is validated. Then I wonder, with the current validation: to what extend TimeFRAME is different from the other packages mentioned by the authors (MixSIAR, simmr)?*

   *Why did the authors not use a model that does consider mixing and loss at the same time to use for data generation? With a limited number of parameters, a system of differential equations can be designed to trace the relevant parameters. Then, you can see to what extend the linearised model used in TimeFRAME is capable of retrieving the relevant parameters in such a case. This would fully validate and allow the authors to quantify the circumstances under which the model is no longer applicable. Making the manuscript far more valuable. Moreover, it would present users with a methodology to come up with their own validation strategies when applying TimeFRAME to their own data.*

   This is a valuable comment that hits upon one of the major challenges in the interpretation of isotopic data. We explored this in more detail in [3] and in the full development version of the package we include examples of how to implement different fractionation equations (eg. Section 4.4.1 of [3]). However, in this manuscript we have focussed on the implementation of Bayesian models for timeseries interpretation and thus focus on the most common version of the mixing and fractionation equation, both due to the length and complexity of the manuscript, and the complications of compiling models at runtime in packages distributed in R. The choice of the reduction model is arbitrary and it is used only to demonstrate the validity of employed statistical tools. The consistency (or so-called 'closure') of the model was demonstrated under a particular assumption regarding the reduction model, however, there is no reason for the time-dependence modelling to lose its consistency when combined with any different reduction mechanism - as long as the self-consistency

is achieved for modelling mixing at individual points in the time series. We hope in a future package version and publication to implement and thoroughly test flexible fractionation equations TimeFRAME, and release the next version of the package in both R and Python. Users who already want to use other fractionation equations in the current version of TimeFRAME will find the necessary tools in the development version of the code, listed in the 'Code and data availability' section.

To clarify this issue in the manuscript, we have added a number of points to further explore this assumption and to support the use of this equation in the current version of TimeFRAME:

- We have added a supplementary section to explore the implications of this assumption:

**Fractionation followed by mixing, or mixing followed by fractionation?**

We simulated $N_2O$ isotopic composition using the endmembers given in Table 1 of the main article to explore differences between the two scenarios (SI Figures 1 and 2):

(a) **Mixing followed by fractionation due to reduction (MR):**

$$\delta_{MR} = f_D\delta_D + f_N\delta_N + \epsilon\ln(r) \tag{1}$$

where $D$ = denitrification, $N$ = nitrification and $r$ = fraction of total $N_2O$ remaining following reduction

(b) **Fractionation due to reduction followed by mixing (RM):**

$$\delta_{RM} = (f_D \times r_D)(\delta_D + \epsilon\ln(r_D)) + f_N\delta_N \tag{2}$$

where $r_D$ is the fraction of $N_2O$ from denitrification remaining after reduction, whereby $r_D = \frac{r}{f_D}$ (SI Figure 1).

In the RM scenario, $r$ cannot be larger than the $f_D$, or $r_D$ would be $> 1$, which implies at least some degree of mixing before fractionation. The $1\sigma$ uncertainty in resulting isotopic composition was estimated using error propagation with the $1\sigma$ uncertainties in $\delta_D$, $\delta_N$ and $\epsilon$ as given in Table 1 of the main article.

Comparing these simulations show that both scenarios deliver the same general trend, whereby isotopic composition increases as the fraction of $N_2O$ remaining following reduction decreases. In both cases, there is a significant difference between the scenarios only when the fraction of $N_2O$ remaining is very low. For $N_2O$, the major uncertainty is contributed by the isotopic endmembers rather than the fractionation model; isotopic endmembers for different source or emission categories are similarly uncertain for other trace gases such as $CH_4$ [1, 9]. This is further shown in the 'boma' case study, where static source apportionment using the MR and RM models gave very similar results (Section 4.3 of the main article). Studies show that a large proportion of the range in endmember values is due to true variability rather than measurement uncertainty, for example due to different rate-limiting steps and microbial enzymes under different conditions, thus even with instrumental development, the endmembers are likely to continue to contribute this level of uncertainty [10] except in specific cases such as pure culture studies.

For TimeFRAME, we opt to use the MR model rather than the RM model. In reality, mixing and fractionation are occurring simultaneously, however to represent this in a model would add an extra level of complexity that would introduce too many degrees of freedom to be constrained with isotopic

[Figure]

Figure 1: $\delta^{15}N^{bulk}$ values simulated across a range of 0 to 1 for the contribution of denitrification to $N_2O$ production (contribution of nitrification = 1 - $f(N_2O$ from denitrification) and 0 to 1 for the fraction of $N_2O$ remaining after consumption. $a$) and $b$) show $\delta^{15}N^{bulk}$ and its uncertainty simulated with SI Eq. 1. $c$) and $d$) show $\delta^{15}N^{bulk}$ and its uncertainty simulated with SI Eq. 2. $e$) and $f$) show the absolute difference in $\delta^{15}N^{bulk}$ between the two simulations and its uncertainty. The dotted region in $e$) indicates where there is no significant difference between the two scenarios.

data timeseries. The MR model is a better approximation of the 'true' situation, because it is clear mixing occurs to some degree in real scenarios. This is illustrated for example by observations of net $N_2O$ uptake, showing $N_2O$ produced by other pathways is consumed in complete denitrification. Moreover, many microbes producing $N_2O$ by denitrification cannot produce nitrous oxide reductase, thus in this case mixing must occur before reduction and Eq. 2 does not apply. Users of TimeFRAME should exercise caution in the interpretation of results when the fraction of $N_2O$ (or other trace gas) remaining is very low, and when other factors suggest that reduction before mixing could be predominant.

- We have added text at L49 referring to this section:

  A detailed discussion of the implications of this assumption is given in SI Section 1.

- We also added further text at L531 referring to this point:

  The dual isotope method results are not significantly different for the MR and RM implementations, supporting the assumptions made in Eq. 3 as the basis for the TimeFRAME package: These models only deliver significantly different results in cases where $N_2O$ reduction is very high (see SI Section 1).

- Regarding the point, to what extent is TimeFRAME different from packages such as MixSIAR and simmr: The primary advantage of TimeFRAME is the ability to deal with timeseries data as well as consumption. Previous packages such as MixSIAR, simmr, and FRAME can only deal with timeseries data to the extent that source contributions are estimated independently for each point in a timeseries, which loses the information included in the temporal autocorrelations between data points. MixSIAR can also only deal with consumption in a very limited sense. We have emphasised these advances at:

  L4: However, there is currently no data analysis package available to solve isotopic production, mixing and consumption problems for timeseries data in a unified manner while accounting for uncertainty in measurements and model parameters as well as temporal autocorrelation between data points and underlying mechanisms.

  L13: Incorporation of temporal information in approaches i-iv) reduced uncertainty and noise compared to the independent model i).

[Figure]

Figure 2: $\delta^{15}N^{SP}$ values simulated across a range of 0 to 1 for the contribution of denitrification to $N_2O$ production (contribution of nitrification = 1 - $f(N_2O$ from denitrification) and 0 to 1 for the fraction of $N_2O$ remaining after consumption. $a)$ and $b)$ show $\delta^{15}N^{SP}$ and its uncertainty simulated with SI Eq. 1. $c)$ and $d)$ show $\delta^{15}N^{SP}$ and its uncertainty simulated with SI Eq. 2. $e)$ and $f)$ show the absolute difference in $\delta^{15}N^{SP}$ between the two simulations and its uncertainty. The dotted region in $e)$ indicates where there is no significant difference between the two scenarios.

L76: TimeFRAME uses one independent time step model in which points in a time series are treated independently, and three classes of model to fully incorporate time series information: i) independent time step models, ii) Gaussian process priors on measurements, iii) Dirichlet-Gaussian process priors, and iv) generalized linear models with spline bases.

- We also emphasised that additional tools are available in the development version of TimeFRAME:

The development version of TimeFRAME, including the different edge scenarios explored in this manuscript as well as tools and examples to assist in the implementation of different fractionation equations, can be accessed at:

2. *Equations 1 and 2 (and therefore also 3 but equation 3 is discussed separately above) are approximations. See for example Chapter 4 Mook, 2000. Mook, 2000 noted that for equation 1 as presented in the manuscript by Harris et al., the induced error is indeed small. Nonetheless, as this paper is the fundamental reference for "TimeFRAME" this should at least be mentioned. Additionally, use "≈" for equations where approximations are made to signal the user that the relation is not exact.*

As the reviewer states, the error induced by this approximation is very small, particularly compared to the error from endmembers, measurements, fractionation model, and the other mentioned sources - therefore we used this commonly employed approximation. We have now added the correct designation for Eqs. 1 and 2:

- ...described using the approximated mixing equation [7, 4]:

$$\delta_{\text{mix}} \approx \sum_{k=1}^{K} f_k \delta_k \tag{3}$$

where $\delta_{\text{mix}}$ is the isotopic composition of a mixture of two or more sources enumerated by $k = 1, ..., K$ with isotopic compositions designated $\delta_k$ and fractional contributions to the mixture designated by $f_k$. This approximation assumes that the light isotope has a much greater concentration than the heavy isotope, which is valid for common trace gases such as $CO_2$, $CH_4$ and $N_2O$.

- ...effect of consumption can be approximated using the Rayleigh equation [6, 7, 4]:

$$\delta_{\text{substr,r}} \approx \delta_{\text{substr,r=1}} + \epsilon \ln(r) \tag{4}$$

3. *As mentioned I'm not fully equipped to judge the validity and correctness of section 2.3. Though I certainly see the relevance of documenting the techniques used in section 2.3, I don't understand it's purpose in the current text. I would be more interested to see a paragraph describing the strength and weaknesses of each of method. This can then be reflected on in the discussion. The technical description can be put in an appendix.*

   As this is a technical note, we find that the detailed methodological description is appropriate; the models described in Section 2.3 form the core of TimeFRAME, and we find that $\sim$0.5 pages per model is not overly long. We see the reviewer's point that it would be good to summarise the strengths and weaknesses of each model. We find that this is done in detail in each of the points of Section 3.3, but it would be good to have a summary as suggested. We have therefore added a subsection at the beginning of the previous Section 3.4 (now Section 4):

   4.1 Model selection and application

   TimeFRAME allows different models to be applied with minimal effort, meaning that data can be analysed with several different model set ups to investigate the robustness of results. The independent time step model does not incorporate timeseries information, thus it is recommended only for datasets with independent measurements. The DGP and spline models both perform well, reproducing the input data values and timeseries properties - the spline model was better able to estimate $r$. All models estimate $f$ of different sources across the full range with similar accuracy, however when the fraction remaining $r$ is very low or high the results show much larger error (Figure 8). This is compounded by the difference between MR and RM models at low values of $r$ (SI Section 1). We therefore recommend users test both DGP and spline models for timeseries data, and take results with caution when these models differ strongly. Estimates of very low fraction remaining should also be treated with caution. Despite these points, we find that TimeFRAME offers a strong improvement on previously available methods: Accounting for information contained within timeseries significantly reduces the uncertainty in estimates of $f$ and $r$, and the package application is simple and fast, and easy to document and reproduce.

4. *Results and Discussion section is long. The results are sometimes difficult to find between discussion paragraphs and also some parts that belong in the methodology section. Would it not be better to separate results and discussion?*

   We agree that the organisation could be improved. We have edited Sections 3.1 and 3.2 in response to the reviewer's comments below. We have also renamed Sections 3.1-3.3 as 'Results' to reflect that these are the more technical part of the results. Section 3.4 is now Section 4 so that users who want an overview of the application of TimeFRAME can locate this section more easily. Finally, we have changed the hierarchy of subsection organisation within the new Results to better reflect the importance of model comparison compared to the prior distribution.

5. *More suggestions*:

   (a) *Section 3.1 is not a result, it is a software design choice. A single paragraph in Section 2 could convey the same information: "We have settled on Stan over JAGS because of performance." Also, another column in table 3 could be used to specify runtime per experiment the give the author an idea of how long each computation takes. Then Table 2 can be removed altogether. (note I also have a question in the minor comments below on this topic)*

We have moved Section 3.1 to the supplementary material, and added a brief overview of the model implementation to the end of the methods. We moved a short relevant section from earlier in the Methods to this part also.

2.5 Model implementation

TimeFRAME is implemented in R [8]. Bayesian modelling in TimeFRAME uses `Stan` for Hamiltonian Monte Carlo sampling (see SI Section 2). TimeFRAME can be installed using the links provided in the *Code and data availability* section. All experiments were run on an Intel Core i9-10900K CPU. The reported run times in SI Section 2 are for a single sampling chain, and in other sections the reported times are the maximum of four simultaneously run chains.

We prefer to leave Table 2 and not incorporate the information into Table 3 - the results and discussion is already detailed, as pointed out by the reviewer, and except for very large datasets all models are sufficiently fast that runtime should not feature in model selection. We therefore prefer not to include the runtime in the model comparison.

(b) *Section 3.2 starts of with the need to test the prior distribution for the fraction remaining $r$. Shouldn't that be part of the methodology?*

We agree with the reviewer - we have moved the first paragraph of Section 3.2 to the Methods, so now only the relevant parts of this discussion are included in the Results.

(c) *Table 3 is part of the methods. I suggest introducing the application of the different models in a separate section after 2.4.1. This section could alternatively benefit from a clear presentation of the different settings for the parameters in the model. (in other words what arguments are passed on to the function in the software package).*

and (combined with following comment):

*Although the technical description is certainly relevant, I miss a bit of an explanation how the user would determine the prior. For example, in Figure 2, if the user were to know a priori that at fi(t = 0) = 1 how is that achieved? When should I use a Gaussian process prior and when a Dirichlet-Gaussian prior? It would be helpful to potential users, what kind of information needs to be defined per model. Throughout the text (For example L 457) recommendations on the usage of the different models can be found. It would be helpful to have a single paragraph (table?) that summarize the recommendations for different applications.*

We hope that the new section 4.1 introduced under point 3 above answers this question, by clarifying choice between the models for application. Regarding parameters: TimeFRAME can be run with only the user data provided, as all other parameters including isotope values for $N_2O$ production and consumption are provided as defaults. This is shown in the package examples, which also illustrate how to provide different inputs if desired. As in all modelling approaches, the choice of parameters is challenging - as now stated in section 4.1, we encourage users to test several configurations and aim to achieve robust results.

(d) *I miss a paragraph discussing the application of TimeFRAME to other trace gas isotope timeseries. They are mentioned in the introduction and abstract (CH4 , CO2 ), but not discussed, this also clearly relates to my major issue. What can the authors see about applications to other isotopes, for example atmospheric CH4 , which has several loss terms in the Stratosphere, with temperature dependent isotope effects? The abstract seems to suggest that this software package is capable of dealing with that.*

The reviewer is correct, this is an omission. The package was particularly designed with $N_2O$ in mind and we have not conducted extensive testing for other trace gases. The current version of TimeFRAME

can deal with as many sources and isotopic dimensions as desired, but only one sink. The model can be solved with one axis of autocorrelation; in the examples here, this is time, however it could also be temperature as in the $CH_4$ example mentioned by the reviewer, or for example WFPS in an incubation experiment for $N_2O$ production. We have clarified this at the end of the new Section 4.1:

The testing here focuses on interpretation of $N_2O$ isotope data to unravel production and consumption pathways. TimeFRAME can also be applied to other scenarios, for example trace gases such as $CO_2$ or $CH_4$, or datasets with many more isotopic dimensions through clumped isotope measurements. The number of sources is indefinite as the model can be extended by the user; however, when the number of sources is larger than the number of isotopic dimensions the model will be poorly constrained. The model can currently only include one consumption pathway applied after mixing - future versions will include more complex set ups, however, the uncertainty in input data currently precludes this level of complexity. The examples shown here use time as the dimension of autocorrelation, as timeseries are the most common kind of data. However, other dimensions could be used, such as temperature in the case of measurements across a gradient of temperature-dependent processes, or soil moisture for a set of incubations across a moisture gradient. TimeFRAME's set up allows simple adaptation to different user-defined mixing and fractionation models and fast and reproducible interpretation of these models.

6. *Minor details*

(a) *L10 here it states " .. production, mixing and consumption .. ", however the rest of the text considers only 2 endmember mixing with a single consumption pathway. What if there are three? or what if there are two loss processes? Please update the abstract, title, and manuscript to be in line with what is presented.*

We have used a classic model with two pathway mixing and fractionation to test the model in this paper, however the user can easily define further sources. This is now mentioned in the text, see point 5d above. This is also shown in the package documentation.

(b) *L18 Consider updating this after addressing my comment on L468.- I think an additional isotope can be very useful in cases where endmembers are significantly different.*

We have added here: ; however, the addition of isotopic dimensions orthogonal to existing information could strongly improve results, for example clumped isotopes. See also the response to point

(c) *"... substrate being consumed before consumption ... " reads a bit odd, simply put " .. the substrate before its consumption .. "*

We have reformulated: initial substrate prior to consumption

(d) *I could not find and access Fischer 2023. So the derivation of this is missing.*

We apologise - this is a MSc thesis that formed the basis of TimeFRAME. The MSc thesis is available through ETHZ's library but challenging to find due to volume. We have added now hosted the thesis directly and added the url to the citation so that it can be easily found, and clarified that this is a MSc thesis:

Fischer, P.: Using Bayesian Mixing Models to Unravel Isotopic Data and Quantify N2O Production and Consumption Pathways (MSc thesis, ETHZ), Msc thesis, ETH Zurich, https://blogs.ethz.ch/eliza-harris-isotopes/timeframe/, 2023.

(e) *L 62 MixSIAR and simmr are mentioned here, but are not reflected on in the discussion. Would it not be informative to include a comparison without a sink? Or at least reflect on those packages? For mixing-only application is there a substantial difference? Does TimeFRAME offer a unique functionality?*

We have added additional information about the unique functionality of TimeFRAME in response to point 1.

(f) *L 79 The remainder of the text suggest that only one fractionation factor can be used?*

One fractionation factor is required for each isotopic dimension, however only one fractionation factor can be used per dimension.

(g) *L 219 Please refer to the original text, as I'm not sure if Fischer 2023 is appropriate here. (potentially other places as well)*

Apologies again that this text could not be found by the reviewer, this was an oversight on our part. As mentioned under 6d, this is an MSc thesis with detailed derivations and/or citations for all equations mentioned here, and we have now made it findable for the reader.

(h) *L 221 Is it necessary for this text to fully explain the DGP process. Would it not be sufficient to simply state that a Dirichlet-Gaussian process is modelled?*

We believe as this is a technical note it is worth fully stating technical information. However, following the reorganization as suggested by the reviewer, we hope it is now easier for the user (who may be less interested in derivation) to find the information needed regarding applications - now section 4 in the revised paper.

(i) *L 272 Why not test with three sources. How well would your model distinguish the bl and b2 sources for oxygen-18? Also, fractionation factors are often variable, for example with a temperature dependence. There is no discussion on the applicability of this model to that specific case.*

The manuscript is focussed on the case of $N_2O$; we find that thorough testing of one case is better than scattered testing of many systems. As the reviewer notes, the manuscript is already long and extensive testing of many other systems is beyond the scope of this study. We hope the new section 4.1 helps better understand the scope and application and we do look forward to extending and testing TimeFRAME for further cases in future.

To clarify, we added to the abstract at L12: We show extensive testing of the four models for the case of $N_2O$ production and consumption in different variations.

(j) *Figure 3. You have only two sources whereas Table 1 lists three. Why is there no test case with all three sources? What can you conclude about more sources?*

We used three sources including the fD pathway to interpret the boma data (Section 3.4.2 in the pre-review manuscript, Section 4.3 in the updated manuscript), however in the absence of ground truth the quality could not be assessed. The model equations will perform similarly for three sources with three isotopes, as for two sources with two isotopes (and analogously with more sources and more isotopes), therefore we did not discuss this testing at all. The test with addition of $\delta^{18}O$ was specifically to examine the case of an overdetermined system, with more isotopic dimensions than pathways as stated at L463, thus added fD to this analysis would not have served this purpose.

(k) *L 304 " ... is a useful metric for to evaluate ... " is bad sentence, please rephrase*

We have fixed this to: Posterior interval coverage is a useful metric to evaluate simulated data from the full Bayesian model

(l) *L 316 and elsewhere, it is recommended that mathematical operators should be written upright Roman. See (Cohen et al., 2007) chapter 4. So the differential d in the integral should be d.*

This has been fixed.

(m) *L 329 The conclusion to use Stan is based on timing tests as presented in Table 2. Yet there seems to be a fundamental difference in the sampling strategies. I assume that those don't matter for your application?*

As stated at L341, the conclusion was based only partly on the timing, which was similar for both models, but primarily because '...the hierarchical model could only be efficiently sampled by Stan'.

(n) *L 347 Doesn't this simply reflect the fact that, for large fractions remaining the effect on isotope measurements is simply so small that it is difficult to see? Especially in the light of the noted uncertainties in Table 1. In other words, if the fractionation constants are well know, and much larger, the change in isotope measurements is much larger and you would be able to see that. Therefore, I think it is strange to attribute this to the nature of the logarithm. In other words, you simply can't resolver very well given the values in Table 1.*

Yes, this is correct, the high uncertainty when $r$ is close to 1 is because of the small fractionation effect. In section 3.2 at L347 we are rather discussing that, because the relationship between $r$ and substrate isotopic composition is non-linear, the prior distribution choice is important or the prior and its uncertainty is not correctly represented. We hope this is made clearer by the reorganisation of this section into Methods and Results, and changes made in response to previous comments regarding uncertainty at low and high $r$, eg. point 3.

(o) *L 370 what about uncertainty in the fractionation factors?*

Apologies for this omission, this is certainly also an important source of uncertainty: The standard deviation between different repetitions however clearly shows that the effect of prior choice is overwhelmed by the variation introduced through the distribution of the sources and consumption, due to the large uncertainty in the source and fractionation factor priors.

(p) *L 420 To what extent does the described experiment here relate to the experiment described in section 3.2*

The experiments are similar but have some small changes as reported, to address the different foci of these subsections. We hope the clarity is now improved as some parts of Section 3.2 were moved to the methods and the results were reorganised to highlight the experiments (point 4 above).

(q) *L 427 This is what I meant with my comment on L 347... Those two descriptions are in conflict. This seems to be the valid analysis.*

These analyses have different foci; first the direct impact prior distribution that is used for $r$, and then the performance of the model more generally across the range of $r$. We hope this is clarified with the responses to the other comments on these sections.

(r) *L 468 "Very little improvement is seen for the estimation off." I think this is to be expected. Looking at table 1, d18O source signatures are very similar for nitrification and denitrification with in there respective uncertainties. I think you would see improvements when adding the third source, fungal denitrification. Why didn't the authors consider that for this experiment?*

As stated in response to point 6j, we were specifically focussing here on an overdetermined system and therefore did not test the fD pathway in this case. To clarify, we have added at L469:

The addition of $\delta^{18}O$ to this model did not strongly improve results, due to large uncertainty in source endmembers and fractionation factors. However, the addition of isotopic dimensions with low uncertainty or strong differences to existing information could improve results, for example clumped isotopes, or $\delta^{18}O$ for determination of fungal denitrification.

Moreover, this information was added to the abstract in brief in response to point 6b.

(s) *L 470 Is this in any case realistic? Natural variability can put a limit on endmember uncertainty reduction through measurements.*

In the case of $N_2O$ production in soils this is not realistic, but for source mixtures like industrial and other sources it could be, and similarly for other isotopic dimensions or systems. It is now clarified that we do not expect this to be the case for $N_2O$ in soils, but are treating this as an experiment:

This experiment is an extremely idealised case and natural variability likely precludes this level of precision in endmembers for microbial $N_2O$ production in soil, however it shows the high potential for improvements in input data to enhance results, and moreover to make results more robust towards model configuration. Currently, the level of uncertainty in direct anthropogenic $N_2O$ and $CH_4$ source endmembers (eg. industrial production, energy and transport emissions) is very high due to the scarcity of measurements [2, 9, 5] - further investigation of the isotopic range of these sources, as well as consideration of endmembers for novel isotopes such as clumped species, may lead to the level of uncertainty reduction required to achieve accurate source partitioning.

**References**

[1] S. Eyer, B. Tuzson, E. Popa, C. van der Veen, T. Roeckmann, W. Brand, R. Fisher, D. Lowry, E. Nisbet, M. Brennwald, E. Harris, L. Emmenegger, H. Fischer, and J. Mohn. Real-time analysis of $\delta$13C- and $\delta$D-CH4 in ambient methane with laser spectroscopy: Method development and first inter-comparison results. *Atmospheric Measurement Techniques*, 9:263–280, 2016.

[2] S. Eyer, B. Tuzson, M. Popa, C. Van Der Veen, T. Röckmann, M. Rothe, W. Brand, R. Fisher, D. Lowry, E. Nisbet, M. Brennwald, E. Harris, C. Zellweger, L. Emmenegger, H. Fischer, and J. Mohn. Real-time analysis of -13C- And -D-CH4 in ambient air with laser spectroscopy: Method development and first intercomparison results. *Atmospheric Measurement Techniques*, 9(1), 2016.

[3] P. Fischer. Renku project: N2O Pathway Analysis, 2023.

[4] P. Fischer. *Using Bayesian Mixing Models to Unravel Isotopic Data and Quantify N2O Production and Consumption Pathways (MSc thesis, ETHZ)*. Msc thesis, ETH Zurich, 2023.

[5] E. Harris, S. Henne, C. Hüglin, C. Zellweger, B. Tuzson, E. Ibraim, L. Emmenegger, and J. Mohn. Tracking nitrous oxide emission processes at a suburban site with semicontinuous, in situ measurements of isotopic composition. *Journal of Geophysical Research - Atmospheres*, 122:1–21, 2017.

[6] A. Mariotti, J. C. Germon, P. Hubert, P. Kaiser, R. Letolle, A. Tardieux, and P. Tardieux. Experimental-determination of Nitrogen Kinetic Isotope Fractionation - Some Principles - Illustration For the Denitrification and Nitrification Processes. *Plant and Soil*, 62(3):413–430, 1981.

[7] N. E. Ostrom, A. Pitt, R. Sutka, P. H. Ostrom, A. S. Grandy, K. M. Huizinga, and G. P. Robertson. Isotopologue effects during N2O reduction in soils and in pure cultures of denitrifiers. *Journal of Geophysical Research - Biogeosciences*, 112(G2):G02005, apr 2007.

[8] R Core Team. *R: A Language and Environment for Statistical Computing*. R Foundation for Statistical Computing, Vienna, Austria, 2017.

[9] T. Röckmann, S. Eyer, C. Van Der Veen, M. Popa, B. Tuzson, G. Monteil, S. Houweling, E. Harris, D. Brunner, H. Fischer, G. Zazzeri, D. Lowry, E. Nisbet, W. Brand, J. Necki, L. Emmenegger, and J. Mohn. In situ observations of the isotopic composition of methane at the Cabauw tall tower site. *Atmospheric Chemistry and Physics*, 16:10469–10487, 2016.

[10] L. Yu, E. Harris, D. Lewicka-Szczebak, and J. Mohn. What can we learn from N2O isotope data? Analytics, processes and modelling. *Rapid Communications in Mass Spectrometry*, 2020.